# The onset of Neoglaciation in Iceland and the 4.2 ka event

Áslaug Geirsdóttir<sup>1</sup>, Gifford H. Miller<sup>1,2</sup>, John T. Andrews<sup>2</sup>, David J. Harning<sup>1,2</sup>, Leif S. Anderson<sup>1,3</sup>

Thor Thordarson<sup>1</sup>

<sup>1</sup>Faculty of Earth Science, University of Iceland
 <sup>2</sup>INSTAAR/Department of Geological Sciences, University of Colorado Boulder
 <sup>3</sup>Department of Earth Sciences, Simon Fraser University, CANADA

Correspondence to: Áslaug Geirsdóttir (age@hi.is)

Abstract. Strong similarities in Holocene climate reconstructions derived from multiple proxies (BSi, TOC,  $\delta^{13}$ C, C/N, MS,  $\delta^{15}$ N) preserved in sediments from both glacial and non-glacial lakes across Iceland indicate a relatively warm early-to-mid Holocene from 10 to 6 ka, overprinted with cold excursions presumably related to meltwater impact on North Atlantic

- circulation until 7.9 ka. Sediment in lakes from glacial catchments indicates their catchments were ice-free during this interval. Statistical treatment of the high-resolution multiproxy paleoclimate lake records shows that despite great variability in catchment characteristics, the records document more or less synchronous abrupt, cold departures as opposed to the smoothly decreasing trend in Northern Hemisphere summer insolation. Although all lake records document a decline in summer temperature through the Holocene consistent with the regular decline in summer insolation, the onset of significant
- summer cooling, occurs ~5 ka in high-elevation interior sites, but is variably later in sites closer to the coast, suggesting some combination of changing ocean currents and sea ice modulate the impact from decreasing summer insolation. The timing of glacier inception during the mid-Holocene is determined by the decent of the Equilibrium Line Altitude (ELA), which is dominated by the evolution of summer temperature as summer insolation declined as well as changes in sea surface temperature for glacial systems particularly in coastal settings. The glacial response to the ELA decline is also highly
- dependent on the local topography. The initial nucleation of Langjökull in the highlands of Iceland starting by ca 5 ka, was followed by a stepwise expansion of both Langjökull and northeast Vatnajökull between 4.5 and 4.0 ka, with a second abrupt expansion ca. 3 ka. However, the initial appearance of Drangajökull in the NW of Iceland was delayed until after 2.5 ka. All lake records reflect abrupt summer temperature and catchment disturbance at about 4.5 ka, statistically indistinguishable from the ~4.2 ka event with a second widespread abrupt disturbance centered on 3.0 ka. Both are intervals of large explosive
- volcanism on Iceland. The most widespread increase in glacier advance, landscape instability, and soil erosion occurred shortly after 2 ka, likely due to a complex combination of increased impact from volcanic activity, cooling climate, and increased sea ice off the coast of Iceland. All lake records indicate a strong decline in temperature ~1.5 ka, culminating during the Little Ice Age between 1300 and 1900 CE when most glaciers reached their maximum dimensions.

#### **1** Introduction

A number of millennial-to-centennial scale proxy-based climate reconstructions provide essential information for understanding the mechanisms behind Holocene climate variability. Although these records are spatially heterogeneous they

- illustrate warmer-than-present conditions during the Holocene Thermal Maximum (HTM, 11-6 ka). In the northern North Atlantic, the HTM was delayed by up to 3000 years compared to the west (Kaufman et al., 2004) likely due to the lingering effect of the residual Laurentide Ice Sheet (LIS) and the Greenland Ice Sheet (GIS), that affected surface energy balance, with variable meltwater fluxes from these ice sheets impacting ocean circulation (Barber et al., 1999; Renssen et al., 2009, 2010; Blaschek and Renssen, 2013; Marcott et al., 2013; Sejrup et al., 2016). Although the primary forcing for the Northern
- Hemisphere's subsequent cooling is the spatially uniform decline in summer insolation (Berger and Loutre, 1991), orbital forcing also led to changes within the internal climate system (e.g., Wanner et al., 2008). On decadal to multi-centennial timescales, climate variability is linked to forcing factors such as solar activity, large tropical volcanic eruptions, responding with internal variability such as the North Atlantic Oscillation (NAO), changes in the Atlantic Meridional Overturning Circulation (AMOC) and its capacity to transport heat in the North Atlantic. Additionally, feedbacks between ocean,
- atmosphere, sea ice and vegetation changes complicate the global response to primary insolation forcing (Trouet et al., 2009; Miller et al., 2012; Lehner et al., 2013; Moreno-Chamarro et al., 2017).

A recent synthesis shows that a general trend of mid Holocene glacier growth in the Northern Hemisphere is related to declining summer temperatures, forced by the orbitally-controlled reduction of summer insolation (Solomina et al., 2015). However, spatially divided compilations based on the Arctic Holocene Transitions (AHT) database (Sundquist et al., 2014)

- indicate heterogeneous responses to the cooling between Arctic regions (Briner et al., 2016; Kaufman et al., 2016; Sejrup et al. 2016). In regard to sea surface and terrestrial summer temperatures for the last 10 ka, the North Atlantic is particularly complex. These records show a strongly uniform pattern related to a direct response to changes in summer insolation, but that the sub-regional differences are apparently related to both the strong influence of the LIS's disintegration and meltwater discharge in the early Holocene and the variability in the strength of the thermohaline circulation throughout the remainder
- of the Holocene (Sejrup et al., 2016).

Iceland lies in the northern North Atlantic, strongly affected by the northward heat transport of the AMOC (Fig. 1), and within the periphery of the strongest contemporary warming (Overland et al., 2016). Paleoclimate research across Iceland shows that the positive summer insolation anomalies of the early Holocene resulted in an "ice-free" Iceland by ~9 ka (Larsen et al., 2012; Striberger et al., 2012; Harning et al., 2016a). Glacier modeling experiments suggest that the observed

disappearance of Langjökull in central Iceland by 9 ka required summer temperature 3°C above the late 20<sup>th</sup> Century average (Flowers et al., 2008), an estimate reinforced by simulations for Drangajökull, an ice cap in northwest Iceland (Anderson et al., 2018). Reconstructions from two high-resolution lake-sediment records in Iceland show that by ~5 ka summertime

cooling had commenced (Geirsdóttir et al., 2013). This cooling was likely intensified in response to lower sea-surface temperatures around Iceland and increased export of Arctic Ocean sea ice (Gudmundsson, 1997; Geirsdóttir et al., 2009a; Larsen et al., 2012). Iceland's ice caps reached maximum dimensions during the Little Ice Age (LIA, ~1300-1900 CE; Larsen et al. 2015, Harning et al., 2016b; Anderson et al., 2018). Despite the monotonic decline in summer insolation, high-

- resolution Icelandic records of environmental change indicate that Iceland's landscape change and ice cap expansion were non-linear (Geirsdóttir et al., 2013).
   In order to understand the non-linear pattern and stepped changes in the mid-to-late Holocene record from Iceland and how regional temperatures evolved in terms of timing and magnitude after the HTM, this paper composites six different
- normalized climate proxies (i.e. mean = 0 ± 1) in high-sedimentation-rate cores from seven lakes in Iceland. By adding five new sediment records to the original two-lake composite of Geirsdóttir et al. (2013), we can more effectively evaluate whether individual site-specific proxy records reflect regional climate change or more local catchment-specific response that are less directly coupled to climate. To reach this goal, we employ a statistical comparison of our proxy reconstructions from the seven lakes in order to quantitatively evaluate their between-lake correlation over time and by inference, their utility as proxies for regional climate. From this treatment we conclude that the primary driver explaining changes in our climate
- proxies was the decline in summer insolation, with changes in ocean circulation around the coast of Iceland and their associated sea surface temperature (SST) modulating primary insolation forcing. Explosive Icelandic volcanism produced important perturbations to lake catchments, often resulting in catchment systems recovering to a different equilibrium state.

#### 2 Study sites

- The temperature and precipitation gradients across Iceland reflect the island's position between cold sea-surface currents from the Arctic and warmer, saltier sea-surface currents from lower latitudes delivered by discrete limbs of the AMOC (Fig. 1). Consequently, climate perturbations that alter the strength, temperature, and/or latitudinal position of these currents should translate to significant changes in the mean terrestrial climate state of Iceland. As an example, sea ice imported and locally formed off the north coast of Iceland during extreme cold events provide a positive feedback on the duration and
- severity of short-term perturbations from volcanic eruptions (Miller et al. 2012; Sicre et al., 2013; Slawinska and Robock, 2018).

Due to Iceland's location astride the North Atlantic Ridge, frequent volcanism influences Iceland's environmental history and is preserved in lake sediment records. The basaltic bedrock generated during the island's volcanic origins is easily erodible, which results in high sedimentation rates within Icelandic lakes, and, hence, the potential for high-resolution

climate records. The frequent production of thick tephra layers have also periodically stripped the landscape of vegetation and triggered intensified soil erosion (cf., Arnalds 2004; Geirsdóttir et al., 2009b; Larsen et al., 2011; Blair et al 2015; Eddudóttir et al., 2017). Clear evidence exists for widespread soil erosion across Iceland during the past thousand years. Even though the prevailing paradigm favors human settlement as the trigger for widespread erosion, a combination of natural climate deterioration and volcanism also likely played a role (e.g., Geirsdóttir et al, 2009b).

The seven lakes studied here (Table 1, Fig. 1, 2) lie on a transect crossing the temperature and precipitation gradient from south to northwest (Fig. 1). Furthermore, they represent a range of catchment styles, including lowland/coastal lakes, highland lakes, non-glacial and glacial lakes, and span elevations between 30 and 540 m asl. The lakes are of varied sizes, from 0.2 to 28.9 km<sup>2</sup>, and lake depths range from 3 to 83 m (Fig. 2, Table 1). In most years, the lakes are ice covered

- from November to April, although early winter thaws and/or late season ice cover may occur. All lakes contain a high-resolution Holocene record with sediment thickness ranging from 2.5 m (SKR) to 20 m (HVT) and occupy glacially scoured bedrock basins. Vestra Gislholtsvatn (VGHV), Arnarvatn Stóra (ARN), Torfdalsvatn (TORF), Haukadalsvatn (HAK) and Skorarvatn (SKR) have not received glacial meltwater since their initial Holocene deglaciation, preserving organic rich sediment throughout the Holocene. In contrast, Hvítárvatn (HVT) and Tröllkonuvatn (TRK) are glacial lakes that have
- captured glacial meltwater when ice margins were within lake catchments following early Holocene deglaciation. Shrubheath characterizes the vegetation around most of the lakes with low-lying plants and mosses dominating the shallow soils. The one exception is VGHV, the southernmost lake, which is predominately surrounded by large planted hayfields today. However, shrub-heath grows near bedrock outcrops on elevated areas around VGHV, whereas fen/mires prefer locations where water can accumulate in lower elevation areas. HAK was submerged by the sea during the last deglaciation (marine
- limit 70 m asl) and the basin became isolated from the sea about 10.6 ka (Geirsdóttir et al. 2009a). Three of the lakes, VGHV, HVT and ARN, lie within the current active volcanic zone of Iceland, and therefore, contain more frequent and thicker tephra layers compared to the other four lakes that all lie distal to the main volcanic zones.

#### **3** Methods

- Regional factors such as volcanism modulate climatic responses to radiative forcing, whereas catchment characteristics may produce different site-specific responses to climate change. Geirsdóttir et al. (2013) composited six different normalized climate proxies in high-sedimentation rate (≥1 m ka<sup>-</sup>1) cores from two lakes, a glacial (HVT) and a non-glacial (HAK) lake. Despite the different catchment-specific processes that characterized each lake's catchment, the composite proxy records show similar stepped changes toward colder states after the mid-Holocene. The common signal in very different catchments
- indicates that the climate proxies in the sediment records reliably reflect past climate change in Iceland. Here we add five new Icelandic lake sediment records and test whether we can replicate the two-lake composite record in other lakes across Iceland. First, we generate a single multi-proxy composite record for each lake by applying the same methodology as in Geirsdóttir et al. (2013). Before calculating each regional mean record, all records were standardized over the whole record/period, which permits reliable comparison of the records with each other.

#### 3.1 Background on the physical and organic proxies and composites

The seven multi-proxy lake composites are based on the average of the normalized values for each of the six environmental proxies; the organic matter (BSi, d<sup>13</sup>C, d<sup>15</sup>N) reflecting biological productivity (mainly diatom growth) and summer warmth, minus the physical proxies (TOC, MS, C/N), which reflect erosional activity and cold and windy seasons within the

catchment. As explained in Geirsdóttir et al. (2009b), TOC is grouped with the physical proxies (catchment instability) in the composites due to the flux of soil carbon into the lake catchments during periods of severe soil erosion occurring during cold, dry and windy seasons. All proxies are given equal weight in each lake's multi-proxy composite (Geirsdóttir et al., 2013). Analyseries software (Paillard et al., 1996) was used to resample each proxy time series to the same 20-year

5 increments. The 20-year interval reflects the minimum resolution within the least resolved dataset used in the analysis (i.e., the BSi record). The C/N ratio was calculated and used to reflect the ratio of terrestrial versus aquatic carbon in the lakes, and hence, the degree of erosion in the lake catchments. The individual proxy data used here have been published previously, and corresponding methods, geochronology, and interpretations are described in the relevant original publications (Table 1).

#### 10 3.2 Correlating lake sediment cores

The abundant tephra layers archived in Icelandic lake sediment generally hold diagnostic geochemical fingerprints, which allow them to serve as key chronostratigraphic markers. These tephra horizons have been dated using historical information and radiocarbon measurements of soil, lacustrine and marine archives. The lake sediment record tephra stratigraphies, together with synchronization of paleomagnetic secular variation between four of the lakes (HVT, HAK, ARN, TORF) and a

15 very well dated marine sediment core off the coast of northern Iceland, provide robust chronologies and minimal age uncertainties over the Holocene (e.g., Jóhannsdóttir, 2007; Stoner et al. 2007; Ólafsdóttir et al 2013; Harning et al., 2018a). These detailed chronologies thus allow for direct comparison of the records from nearby lakes to develop regional syntheses of climate history (Geirsdóttir et al., 2013). The age model for each lake was constructed by fitting control points with a smoothed spline using the CLAM code (Blaauw, 2010). All ages in the text are reported as calendar years prior to 2000 AD 20 (b2k).

#### 3.3 Statistical analyses

Our primary goal when reconstructing Holocene climate evolution is to test whether changes on decade-to-century scales are regionally coherent (e.g., Geirsdóttir et al., 2013). This effort requires continuous records at high resolution from different

- 25 catchments where consideration is given to the large number of highly resolved climate proxies derived from each of the seven lake sediment records. We are primarily interested in the agreement or otherwise of the trends in the data. In order to compare both within lake and between lake variables we normalized the resampled data (n = 3478) so that: 1) all variables had equal weight (mean = 0 ± 1), and 2) differences between lakes, possibly due to climatic gradients, are clarified. In order to evaluate the linkages between the lake proxies we applied R-Mode factor analysis to the 6 x 3478 data matrix (Davis,
- 30 1986; Aabel, 2016) to evaluate the proxy records as indexes of regional climate change, rather than site-specific environmental conditions decoupled from regional climatic gradients. Factor analysis extracts the dominant signal (1st PCA), and the explained variance plus each lakes factor scores gives a measure of how the different sites are associated to this main signal, hence allowing us to evaluate the importance of an individual proxy as a climate recorder.

5

#### 4 Results and interpretation

#### 4.1 Proxy measurements and the composite Holocene records

The multi-proxy composites for each lake reduce proxy-specific signals within each lake, while amplifying those signals that are recorded by most or all proxies (Fig. 3a). By isolating each individual lake composite record, the signal of catchment specific processes within each lake record is preserved, which validates our comparison between different catchments to test for a regional. Iceland-wide signal.

#### 4.2 Results from Factor analyses

We employ the statistical comparison between our proxy-based reconstructions from the lakes in order to quantitatively 10 evaluate their correlation over time and by inference, their utility as proxies for regional climate. The results (Table 2) 10 indicate that the 1<sup>st</sup> factor explains 56.9% of the variance and has high loadings (associations) with 5 of the proxies. MS and 1<sup>s</sup>N are positively loaded on this axis whereas TC, BSi, and d<sup>13</sup>C have strong negative loadings. Communalities, a measure of shared information (Davis, 1986; Aabel, 2016), is high for all 6 proxies with d<sup>15</sup>N having the highest unique (noise) rating. An additional 19.5% of the variance is explained by the 2<sup>nd</sup> factor, which is largely a measure of the C/N ratio.

15 Varimax factor analysis (Table 2) does not result in any significant changes in the rankings or sign of the proxies, except for BSi, which now has the strongest loading on factor 1. The first two factors explain 76.4% of the data indicating that most of the variance in the dataset may be explained by these factors and that the seven lake sediment records are showing a similar signal.

A plot of the factor scores (Fig. 4), labeled for each of the 7 lakes, indicates a clear environmental gradient along 20 the 1<sup>st</sup> factor axis and a less distinct, but still clear, division on the 2<sup>nd</sup> factor axis, for example between HVT and HAK (Fig. 4). These distinct clusters reflect the combined scaling of each 3478 measurements per proxy.

The similarities in timing and direction in variations of climate proxies preserved in all seven lakes composites, our robust age models, and the communalities (Table 2), enable us to consider all seven composite lake records and the production of a single time series similar in general appearance to the two-lake composite shown in Geirsdóttir et al. (2013)

- 25 (Fig. 5a). The combined composite of all seven lakes (catchments) diminishes the site-specific signals preserved in each lake composite while amplifying those signals that are common in all catchments. Compared to our previous composite (Geirsdóttir et al., 2013), the 7-lake composite is near identical indicating that all lakes experienced similar climatic histories with relatively minor superimposed catchment specific processes (Fig. 5a). Furthermore, the results from the factor analyses suggest that a simple composite of just the BSi as a measure of change in relative spring/summer warmth (e.g., Geirsdóttir et al.)
- 30 al., 2009b) and C/N as a measure of cold or erosional activity could, along with the composites, serve to increase our understanding of the evolution of Holocene climate in Iceland and disentangle the catchment responses to climate change and the temperature forcing. A comparison of such reconstructions with the composite of all proxies/all lakes is shown in Figure 5b and 5c.

#### **5** Discussion

## 5.1 The demise and growth of the Icelandic glaciers during the Holocene

- 5 The overall (first-order) trend of the BSi composite reflects the orbital cycle of summer insolation across the Northern Hemisphere after peak insolation ~11 ka (Fig. 5b,c). The presence of the 10 ka Grímsvötn tephra series (including the so-called Saksunarvatn tephra) in all of the seven lakes and other high-elevation sites across much of Iceland (Stötter et al., 1999; Caseldine et al., 2003; Jóhannsdóttir, 2007; Geirsdóttir et al., 2009a; Larsen et al., 2012; Harning et al., 2018a) demonstrates that the highlands were mostly ice-free by the time of the eruption ~10.3 ka, with most of the Iceland Ice Sheet gone from the interior before ~9 ka (Larsen et al., 2012, Harning et al., 2016a, 2018b; Striberger et al., 2012; Gunnarson,
- 2017).

Two of our lakes, HVT and TRK, are currently glacial lakes and track the expansion of ice caps that deliver glacial meltwater from Langjökull and Drangajökull, respectively (Fig. 2, 6), only when they attain sufficient dimensions. Climate proxies in these lake sediments record the growth and demise of an upstream glacier (e.g., Briner et al., 2010; Larsen et al.,

- 15 2012; Harning et al., 2018b). Both lake sediment records demonstrate that glacier ice disappeared from their catchments prior to 9 ka (Larsen et al., 2012; Harning et al., 2016b, 2018b). Furthermore, detailed sedimentological analyses of HVT indicate that the lake did not receive glacial meltwater between 7.9 and 5.5 ka (Larsen et al., 2012). Modeling experiments show that the demise of Langjökull and Drangajökull during the early Holocene required summer temperatures to rise ~3°C above the late 20<sup>th</sup> Century average at this time (Flowers et al., 2008; Anderson et al., 2018).
- Superimposed on the first-order decrease of BSi-inferred relative temperature are apparent step changes at ~5.5 ka, ~4.5 ka, ~3 ka and ~1.5 ka, with the lowest temperatures culminating during the LIA between 0.7 and 0.1 ka. Low BSi values coinciding with high and increasing C/N values at these steps suggest an increase in the proportion of terrestrial organic matter delivered to the lakes (soil erosion) relative to primary aquatic organic matter during cold times (Fig. 5b, c). The fastest rate of change occurs after 1.5 ka when the BSi values reflect the continuous decline in primary productivity at the same time that C/N values persistently increase, reflecting the catchment response to climate change (Fig. 5c, 6).

Sedimentological analyses of HVT and its BSi record show that the first abrupt change of diminished biological productivity at 5.5 ka reflects a change toward cooler conditions (Larsen et al., 2011; Fig. 3a). The hydrologically coupled ice-sheet model employed to simulate the evolution of Langjökull through the Holocene captures the inception of the modern ice cap prior to 5 ka (Flowers et al. 2008), concurrent with the first abrupt change in the BSi record (Fig. 6a). The

30 second distinct change in HVT at 4.5 ka towards cooler climate is not only reflected in a suppressed algal productivity and increased landscape instability but a shift from weakly stratified to finely laminated sediments signaling the onset of a glacial-lacustrine-dominated HVT catchment. The H4 tephra layer from the explosive Hekla eruption dated to 4.2 ka is found in five out of seven lakes and marks the culmination of this second change towards cooler climate that started around 4.5 ka. At 3 ka HVT developed the first distinctly varved sediment record, reflecting increased glacier activity in the

catchment (Larsen et al., 2011). This coincides with the eruption of the Hekla volcano at 3 ka. The H3 and H4 eruptions were the largest silicic explosive volcanic eruptions of the Holocene in Iceland. The impact of the tephra on the landscape in either case is unambiguous and may thus be one representation of amplified catchment response to the declining summer insolation and contemporaneous cooling that may have resulted in further growth of Langjökull. The fastest rate of decline in

- 5 algal productivity (BSi in HVT) and the most apparent glacier advance began between 1.8 and 1.5 ka culminating in the LIA (Fig. 6a). This pattern is supported by the numerical simulations, which suggest Langjökull attained its maximum volume during the Little Ice Age; first around 1840 CE and then around 1890 CE when estimated temperature were 1.5°C below the 1960-1990 average (Flowers et al., 2007) (Fig. 6c).
- Unlike HVT, glacier ice did not reach Tröllkonuvatn's catchment until 1 ka, as illustrated by the lack of meltwater-10 derived clastic sediment in the lake between 9 and 1 ka (Harning et al., 2018b). However, Drangajökull was actively expanding into another threshold lake catchment on the current southeastern margin by at least 2.3 ka (Harning et al., 2016a). Further evidence for late Holocene advances of Drangajökull come from <sup>14</sup>C-dates on dead vegetation emerging from the currently receding northern and southern ice margins, which define the timing of persistent expansion of ice at these locations (Harning et al., 2016a, 2018b). The threshold lake records combined with <sup>14</sup>C-dated emerging dead vegetation
- 15 define five periods of increasing ice cap dimensions at ~2.3, 1.8, 1.4, 1 and 0.5 ka, where the final two ice margin advances (1 and 0.5 ka) are interpreted from Tröllkonuvatn's sediment record. The numerical simulations by Anderson et al. (2018) supports Drangajökull's maximum expansions between 0.5 and 0.3 ka with temperatures likely 1.0 to 1.2°C lower than the 1960-1990 reference temperature.
- Striberger et al. (2012) studied the sediments of the glacier-fed lake Lögurinn in eastern Iceland to infer Holocene meltwater variability of Eyjabakkajökull, a surge type outlet glacier of NE Vatnajökull (Fig. 2, 6b). The results show that Eyjabakkajökull had ceased to deliver glacial meltwater to the lake by 9.0 ka and that the glacier's dimensions were considerably smaller relative to today at this time. The return of glacial meltwater into the lake at ~4.4 ka indicates the regrowth of Eyjabakkajökull into lake Lögurinn's catchment, which suggests an almost 5000 years long glacier-free period along NE Vatnajökull during the early-to-mid Holocene. Subsequent to 4.4 ka, the lake Lögurinn BSi record reflects a marked and continuous decrease in the aquatic productivity through to the LIA, which suggests that Eyjabakkajökull attained
- 25 marked and continuous decrease in the aquatic productivity through to the LIA, which suggests that Eyjabakkajokull attaine its maximum Holocene extension at this time as well (Striberger et al., 2012).

The results from the Factor Analyses illustrate a clear gradient along the  $1^{st}$  factor axis (BSi) and a less distinct, but still clear, division on the  $2^{nd}$  factor axis (C/N). The fact that this trend is especially evident between HVT, the glacial lake in the highlands, and HAK, the non-glacial coastal lake in western Iceland, supports our previous interpretation (e.g.,

30 Geirsdóttir et al., 2013) that these two lakes form two end members of combined proxy responses to the mid-to-late Holocene forcing toward cooler summers. Great similarities occur between the two highland lakes (HVT and ARN), which both lie within the volcanic zone of Iceland on either side of Langjökull (Fig. 2), although only HVT is currently affected by the glacier (Larsen et al., 2012, Gunnarson, 2017). Importantly, both records show similar timing and direction of the most distinct perturbations throughout the mid to late Holocene (Fig. 3a, b). Similarly, the coastal lakes HAK, TORF, and SKR

share common features with more suppressed punctuations than the highland lakes, perhaps indicating greater impact from the surrounding sea surface temperature and its characteristic thermal inertia. This is supported by the similarity of these records to diatom-based SSTs from the shelf north of Iceland, particularly in regard to how more pronounced cooling is delayed until the late Holocene (Fig. 7; Jiang et al., 2015). TRK, although also a glacial lake, shows more similarity to the

- 5 coastal lakes than glacial lake HVT (Fig. 3a, b) likely due to its shared coastal location outside the active volcanic zone with minimal influence from most volcanic eruptions (Harning et al. 2018b). However, the one lake most affected by volcanism in Iceland is VGHV in the southern lowlands of Iceland (Blair et al., 2015), and thus, seems to form its own group in the Factor Analyses (Fig. 3a, b and Fig. 4).
- An explanation for the step-shifts in the BSi (and other climate proxies) may in part be related to perturbations from volcanism, either Icelandic or a combination of local and tropical volcanism, both of which are likely to produce brief cold summers resulting in catchment instability. But with all climate proxies (all composites) exhibiting severe perturbations at the same time (although of variable magnitude) and their failure to return to pre-perturbation values suggests a changed relation between the landscape and/or climate following each perturbation. The different response of the highland/glacier record compared to the coastal lake records suggests differences in the response of catchment specific processes to the
- orbitally forced summer temperature decline, which in turn depends on geographic location and the impact of the surrounding water (Fig. 6). Although the lake sediment record from the southern lowlands (VGHV) deviates from this, it records brief proxy perturbations following major Icelandic eruptions, but shows limited sustained instability (Blair et al., 2015). The behavior of VHGV's record may suggest the influence of intermittent volcanic activity on the proxies against a backdrop of generally consistently "warm" climate along the southern lowlands induced by the Irminger Current (IC, Fig. 1)
- flowing from east to west south of Iceland. On the other hand, the coastal lakes in northwest Iceland may be affected by variations in SST's of the cooler and more variable North Iceland Irminger Current (NIIC, Fig. 1). However, noteworthy here is the synchronous and accelerated decline after 1.5 ka found within all lake records.

#### 5.2 The "onset" of Neoglaciation

- The time interval known as "Neoglaciation" was defined by Porter and Denton (1967) as "the climatic episode characterized by rebirth and/or growth of glaciers following maximum shrinkage during the Hypsithermal [now HTM] interval". Porter (2000) noted that Neoglaciation is a geologic-climate unit based on physical geological evidence of glacier expansion, and that palynological evidence of climate change was excluded from the definition. The use of the phrase "onset of Neoglaciation" in the literature has been broader than the original definition and is commonly attributed to the first apparent indication of lowered temperature or increased rate of summer cooling rather than simply the renucleation and/or expansion
- of glaciers. Following Porter's (2000) definition, the onset of Neoglaciation in Iceland based on our lake records occurre

Following Porter's (2000) definition, the onset of Neoglaciation in Iceland based on our lake records occurred before 5.0 ka for Langjökull, although the initial growth of Drangajökull occurred much later, ~2.3 ka. This indicates that the spatio-temporal nucleation of glaciers in Iceland was asynchronous likely reflecting the relation between the regional ELA

and topography. The nature of the topography (i.e., the hypsometry) controls how quickly the glacier will expand after the ELA intersects the topography. Because the topography under Drangajökull is a plateau-like landscape with a large area at its highest elevation, Drangajökull will grow quickly once the ELA intercepts the topography (Anderson et al., 2018).

- The current ELA pattern reflects reflect the patterns and temperature and precipitation across Iceland. Temperature differences from the south to the northwest reflect the prevailing wind direction and ocean surface temperature. Precipitation also impacts ELAs across Iceland but varies primarily due to the interaction of local topography with prevailing winds (Crochet et al., 2007). Because Icelandic glaciers are most sensitive to temperature, we expect the timing of glacier inception to be controlled by the rate at which Holocene temperatures decline following the HTM and the unique topographic setting of each of the current Icelandic ice caps.
- Changes in the ELA at present are most sensitive to summer temperature, and assuming the same holds for most of the Holocene, we apply a simple approach to define the "onset" of Neoglaciation in Iceland by comparing the Holocene evolution of BSi (a measure of variations in summer temperature) and C/N (a measure of variations in catchment stability, which independently tracks summer temperature, modulated by volcanism) in the coastal lakes (SKR, TRK, TORF, HAK) and the highland lakes (HVT, ARN) (Fig. 7). Although the multi-proxy lake records document complex changes in terrestrial
- climate and glacier fluctuations in Iceland during the mid-to-late Holocene, coherent patterns of change are apparent. Based on glacier nucleation, our records show that initiation of Neoglacial cooling took place in the highlands of Iceland (Langjökull) around 5.5 ka, where the current ELA is ca 1170 m (Fig. 6a, 7). The rate of glacier growth likely increased between 4.5 ka and 4.0 ka when NE Vatnajökull (ELA 1320 m) nucleated (Fig. 6b), and continued near ~2.5 ka, when Drangajökull nucleated (ELA 675 m) (Fig. 6c). Consistent with this scenario is the distinct first-order cooling trend apparent
- in all seven lakes, including the combined highland (HVT, ARN) and the coastal (SKR, TRK, TORF, HAK) lake composites (Fig. 7). The proxies in the highland lakes, HVT and ARN, show a more abrupt and greater response to the temperature lowering, most likely due to a greater impact from catchment specific processes (volcanism and/or glacial activity), whereas the coastal lakes show a more subdued response, likely reflecting the moderating effect from SSTs (Fig.7). These lakes show not only striking similarities with the diatom-based SST record from the shelf north of Iceland (Jiang et al., 2015) but also
- with the IP<sub>25</sub>-based sea-ice reconstruction (Cabedo-Sanz et al., 2016) and the ice-rafted debris (IRD) record based on quartz grain counts (Moros et al., 2006) (Fig. 7), both off the north coast of Iceland. The IP<sub>25</sub>-based sea ice reconstructions starts to rise from a background state beginning at 8 ka at ~ 5 ka, and intensifying after 4.5 ka, broadly in line with the decreased abundance of planktic foraminifera and lowering of SST (Jiang et al., 2015) (Fig.7). Further increases in drift ice were evident during the late Holocene after ca 3.3 ka, with maximum sea ice after ca 1.0 ka and during the Little Ice Age (Moros
- et al., 2006; Cabedo-Sanz et al., 2016). The intensification around 4.5 ka seen in our lake records is in line with increased strength of cold and fresh Polar Water via the East Greenland Current (EGC, Fig. 1) at 4.5 ka in the northern North Atlantic inferred from the decreased abundance of planktic foraminifera (e.g., Andersen et al., 2004; Jennings et al., 2011; Ólafsdóttir et al., 2010; Kristjánsdóttir et al., 2016; Perner et al., 2015, 2016).

A diatom-based SST reconstruction from a core retrieved from the Iceland Basin south of Iceland shows