# Peer review of "The onset of Neoglaciation in Iceland and the 4.2 ka event"

_Climate of the Past, 2018_

## Referee Comment (RC1) · Anonymous Referee #1 · 19 Nov 2018

The overall quality of the discussion paper ("general comments") The paper summarizes six normalized series of sediment proxies (BSi, TOC, $\delta13C$, C/N, MS, $\delta15N$) from seven lakes with high accumulation rate in Iceland. This way the authors demonstrated the general trends in summer temperature changes and glacier fluctuations in the region. They also discussed potential drivers of these variations, including the orbital forcing, explaining the long-term trend of the holocene summer temperature changes, superimposed with higher frequency perturbations such as local volcanic eruptions and sea ice variability. I really liked the approach: the multi-proxy and multi-lake together make it possible to minimize the individual specificity of each of the lakes and allow the identification of the regional patterns. This has already been done for two lakes earlier, but now the aggregated data is from seven lakes of different origin, located in different

landscape zones and with different types of sediment accumulation. It is important that the signal coincides in many ways and allows to make all lakes-composite. Obviously, all the lake sediment series are well dated and the 14C dates are cross-referenced with tephras, so there should be no questions about the chronology of events. The interpretation of the onset of the Neoglaciation in Iceland and explanations from the point of view of forcing also sound rather convincing to me. The goal of the paper is clear and well defined. The paper is well designed and written, illustrated by informative figures and tables. The methodology is valid and correct. Overall, the paper is of value for a large scientific audience. I think that after very minor corrections the paper is ready for publication.

Individual scientific questions/issues ("specific comments") It seemed to me that it was not completely obvious why these six proxies were chosen. Can you please specify the processes of the selection of proxies. Three of them (BSi, d13C, d15N) reflect bioproductivity and summer temperature, and other three (TOC, MS, C / N) - erosion activity during the cold period. It is curious that TOC refers to the cold period. The authors argue that the TOC increases due to soil erosion occurring during the cold, dry and windy seasons (Geirsdóttir et al., 2009b). However, the logic might be different: more organic material is transported in the lakes due to more intensive precipitation and snow melt. I would suggest to expand a little this part and explain the mechanisms.

A few interesting questions remained outside the scope of the paper, although they might be interesting for the reader.

It would be helpful to compare the reconstruction provided in this paper with those based on pollen analyses. Are they coherent? The dates of moraines are not mentioned. Do they agree with the sediment records? The modern warming is not manifested in the records (see fig. 5c). What is going on nowadays in this region? I think a few words on the current climate trends (including seasonality)in the region are necessary. The Medieval Climatic Anomaly is not mentioned and discussed here. What is your opinion on this? All lake records indicate a strong decline in temperature ∼1.5 ka.

Can you suggest an explanation for the absence of the MCA in this region unlike some other areas in the North Atlantic? I am not sure why the authors limit themselves by the Neoglacial time having the complete Holocene records? I would suggest to reconsider the title. They state that the 4.2 ka event is undistinguishable in the period 4.0 - 4.5 ka, so why it should be the focus and mentioned even in the title? Technical corrections Ðǎ 10 line 4 The current ELA pattern reflects reflect the patterns and temperature and precipitation across Iceland – please edit

―――――――――――――――――

---

## Referee Comment (RC2) · Anonymous Referee #2 · 1 Dec 2018

Summary comments:

This paper focuses on the paleolimnological evidence for episodic cooling on Iceland ca. 4,200 yrs ago, a time associated with numerous other climate changes around the world, and particularly in the North Atlantic. The manuscript synthesizes previously published data from the same group of authors using factor analysis to examine commonality in the signals among the various records, across lake types and proxy types. The paper is an extension of a previous analysis by Geirsdottir and others, but with additional sediment records included. The study is inherently valuable because it brings together a large body of work produced by a single research group during the past decade or so, the records are of high quality, and the age models are good. I think the results will be eventually suitable for publication in CoP, but I do have some

questions and comments that I would like the authors to address. Many of my comments seek clarification of points that are being made in the discussion that I think are too vague and more transparency concerning the basis for determining ages of glacial inception. In a bigger sense, though, I think the manuscript would be improved if it had a more thorough discussion of all the stepwise cooling events that are revealed by the sediment record synthesis, not only the cooling event around 4.2ka.

Reviewer comments:

1. The title of the paper includes the phrase "onset of neoglaciation," and I appreciate that the authors include a discussion of the origin of this term, but when do they believe this onset began in Iceland? It seems this should be a fundamental conclusion of the paper if it is so prominent in the title. The discussion leads one to understand that neoglacial inception was catchment-specific across Iceland, and this is to be expected. Yet, this seems at odds with the conclusion that the neoglacial began at ~5ka (which really comes from Larsen et al., 2012). I don't follow the logic used to determine this timing. Why are the BSi records shown in Fig 6 and 7 used to justify this timing for glacial inception? Do the blue bars on this figure denote onset of minreogenic input to the lakes from glacial erosion? I have either missed it in the discussion, or it could use further explanation. As it stands, I do not find that a neoglacial onset of 5.5ka can be concluded from the records shown. What provides the basis for this?

2. Section 5.2 is called: The onset of neoglaciation. Section 5.3 is called: The onset of neoglaciation in the circum North Atlantic and the 4.2 ka event. I think the authors intended to have a greater discussion of the North Atlantic patterns of / evidence of neoglacial inception in different regions in section 5.3, but this never materializes. It probably should, because there is ample evidence from many parts of the North Atlantic. This discussion should either be expanded or the section title changed to reflect the content. The discussion of neoglaciation in the circum North Atlantic seems too brief. It basically stops at noting that records appear to be driven by monotonic insolation forcing. This is a shame, because there are other studies that previously showed

stepwise cooling in the Holocene (including at 4.2ka) and that provide support for the authors' interpretations. These should be discussed/cited.

3. I understand that this is for a special issue of CoP concerning climate changes that took place around 4.2ka and that this motivation has steered the direction of the discussion of the datasets being synthesized in this manuscript. However, it seems like a missed opportunity to focus only on this single event. Doing so implies to the reader that the changes in the records observed at 4.2ka are somehow bigger, more abrupt, or different in some way from the many other abrupt changes seen in these records. What about the 6.5ka, 5.5ka, 3.0ka, and 1.5ka Events? These all stand out as equally important and noteworthy in the 7-lake all proxies record. The fact that the high latitude North Atlantic cooled through the mid-late Holocene in a stepwise manner is very important. I think a section that discusses the other abrupt cooling steps and their relationships to potential forcing factors, oceanographic changes, etc, as has been done for the changes between 4-4.5ka, would be very valuable.

4. There is an implication running through his manuscript that the climate perturbations seen in these records are due to climate cooling episodes caused by volcanism. This is possibly true. But I think I think this hypothesis should be given it's own discussion rather than being inserted here and there throughout the manuscript. For example, the first line in the method section presupposes that volcanoes are the primary climate forcing responsible for the signals in the proxies – I don't really think this is an appropriate place in the manuscript to insert this concept. Something that strikes me as particularly confusing is that one of the conclusions of the paper is that the Hekla4 eruption, although coincident with the 4.2 event, could not be responsible for the climate cooling because of low SO2 in the eruption. Yet, there is wording throughout the manuscript that leads a reader to believe that volcanism was responsible for climate cooling events observed in the record. I think that a section that specifically focuses on he role of volcanoes on climate and on landscape dynamics would be very useful.

5. The authors interpret C/N and TOC as both directly related to catchment erosion

(pg5 lines 1,2). This may be the case. If so, it can be easily tested by comparing the two data sets. Do TOC and C/N carry the same signal within each lake? This would support the interpretation. How do they correlate within each record? I ask because it seems these signals could be much more complicated. It isn't clear to me how colder conditions lead to greater soil erosion. Is it because there is less vegetation during cold times? Or is it because there is greater glacial erosion? That shouldn't matter in the non-glaciated catchments. Moreover, when BSi decreases shouldn't the in-lake organic productivity also go down, leading to higher C/N even in the absence of changes in terrestrial input? When BSi increases due to increased productivity, won't TOC go down due to dilution of the sediments by BSi, even in the absence of changes in soil erosion? It seems like mass accumulation rates are needed to consider these proxies independently, particularly in the glaciated catchments where changes in glacially derived material is likely the primary control on all of these other measured proxies (as %).

6. Ideally, a synthesis of various records would include error bounds that propagate the uncertainty in the age models of each sediment core. I know the age models are quite good in these records, due to the abundant tephra layers, but correlating the records of lakes to within a couple hundred years is still quite challenging. The correlation uncertainties change as a function of distance from age control points, and the authors have already calculated the age uncertainties for each sediment record using BACON. These should really now be used to propagate these uncertainties into the "all proxy" composites. The abrupt changes are very evident and I do not doubt them and I think they will remain a robust feature, but this uncertainty analysis would be useful.

7. One thing that is unclear to me in the manuscript at times is what is meant by "volcanic impact" on a catchment. It seems as though this is sometimes referring to the indirect impact of volcanism via its impact on climate, and at other times is referring to the actual physical impact on a specific catchment stemming from local volcanism. One example of this is line 22 on page 10, where the authors refer to the greater impact

on HVT and ARN (relative to the other lakes) from catchment-specific processes, including volcanism. I can't tell whether this is implying that volcanism leads to changes in catchment erosion independent of climate that then obscures the climate records, or if the point is that these catchments respond differently to external climate forcing because they are more continental than the coastal sites and a given volcanic eruption impacts them more. I believe that this could use clarification. A dedicated section about the impacts of volcanism on lake records would probably be very useful in clarifying the authors' meaning.

8. Page 7, line 26: "Sedimentological analyses of HVT...toward cooler conditions..." Fig. 3a is referenced here, but there is no BSi record on that figure and the "all proxy record" from HVT shown on Fig 3 actually shows a first cooling step at 6.5ka, not 5.5ka. Is there some other information being used to assign the 5.5ka step as the neoglacial inception? Perhaps mag susceptibility, grain size, or minerogenic content? Especially because "neoglacial onset" is in the title of the paper, it seems that being as clear as possible about the underlying evidence is important. Also, is this change in BSi actually diminished biological productivity, or driven more by the dilution impact of renewed input of glacially derived sediments to the lake? Seems the latter is a better indication of glacier inception.

9. Page 8, line 2-4: "The impact of the tephra on the landscape in either case is unambiguous..." What is the "unambiguous impact" of the tephra on the landscape? There is an implication here that the tephra somehow impacts the catchment response to climate forcing, or maybe confounds the proxies in the lakes of those catchments such that they do not represent climate when there is tephra in the catchment - but exactly how this works and the impact on the proxy interpretations is never discussed. I'd like a more detailed discussion of these impacts.

10. Line 24 page 11: The final words here are "supporting our conclusions," but it is not clear what is meant. I think the authors are saying that other studies that have either documented or speculated about abrupt changes in ocean currents during the

past 2ka support their interpretation that the lake sediments document abrupt cooling events. This isn't really true, first of all; but secondly the wording is too vague and I don't understand what exactly is being linked between these cited papers that have interpretations about the internal feedbacks of the North Atlantic and the conclusions of this paper, which at this point in the paper have not yet been reached, or at least are still a bit vague.

11. Conclusions: There are 6 conclusions of the paper provided as a bulleted list. Some of these are not actually conclusions that can be drawn from the results presented here, and are better described as discussion points than conclusions. Some are not necessarily supported by the data presented. I would ask the authors to be more specific about their conclusions, remove those that aren't really conclusions from this study, and to provide them as a narrative rather than a list, so they can explain/summarize. 1. ELA intercepted Langjokull at 5ka. There isn't evidence for this presented in this paper. 2. This conclusion is saying that the Holocene cooling on Iceland happened in a stepwise manner, which I think is a reasonable conclusion. However, based on the records presented, the first cooling event happened well before 5ka. 3. Stepwise cooling requires internal feedbacks, which possibly involve ocean dynamics. Reasonable conclusion. 4. I think this conclusion is that Hekla 4 eruption did not cause the cooling associated with the 4.2 event, even though they are contemporaneous. Does this conclusion really stem from the results presented in this study? 5. This is about sea ice expansion during neoglaciation, but this isn't a conclusion from this Iceland lake synthesis. 6. Ocean circulation influenced climate on Iceland. This can be a conclusion based on the comparison of the Iceland records with some marine records, but the conclusion should be much more specific than this. Clearly, ocean dynamics impact Iceland's temperature – isn't there more than can be concluded about how and when?

12. Tables: Data tables are incomplete. Table 1 – Can easily delineate and measure the catchment area of the two lakes that aren't included using readily available maps.

[Figure]

Table 1 – title says core description, but the table contains only lake descriptions.

13. There are some typos throughout that the author's should look out for. But one in particular that spellcheck won't pick up on is in the Abstract, where is says decent instead of descent.

───────────────────────────

---

## Author Response (AR1)

Dear Referee

We would like to thank the reviewer for the constructive comments and suggestions that help clarify some issues in this manuscript and improve it. Below we address all of this reviewer's comments and concerns to the best of our ability. Corresponding revisions will be made to the manuscript.

Comment 1 - Individual scientific questions/issues ("specific comments") It seemed to me that it was not completely obvious why these six proxies were chosen. Can you please specify the processes of the selection of proxies. Three of them (BSi, d13C, d15N) reflect bioproductivity and summer temperature, and other three (TOC, MS, C/N)

[Figure]

– erosion activity during the cold period. It is curious that TOC refers to the cold period.

Response: As pointed out by the reviewer, this paper applies the same proxies selected and used by Geirsdóttir et al. (2013). In our 2013 paper, we do explain and describe the selection rationale and interpretation of individual proxies in detail. We feel here that by referring to that paper, we can exclude similar discussions in this paper. However, we have now made corresponding revision to the manuscript where we have added a brief discussion of the reasoning for using these proxies and their interpretation. For a detailed explanation, we direct the reader to the 2013 paper. To specifically address the reviewer's question about TOC, we suggest that TOC in the sediment is a product of both production and transport terms (among other factors). The production term is increased during warm periods due to increased plant growth, but transport is minimized as organic material in the catchment accumulates (and remains sequestered) in soils. During cold periods, even though the production term is minimized, the transport of previously accumulated organic matter from eroding soils contributes a large influx of OC to the lake sediment. This more than compensates for any decrease in productivity due to shorter growing seasons and leads to a net increase of sediment TOC during cold periods.

Comment 2 - The authors argue that the TOC increases due to soil erosion occurring during the cold, dry and windy seasons (Geirsdóttir et al., 2009b). However, the logic might be different: more organic material is transported in the lakes due to more intensive precipitation and snow melt. I would suggest to expand a little this part and explain the mechanisms.

Response: Please see the response to comment 1 for an explanation of the mechanisms that lead to increased TOC during cold periods. The soils of Iceland lack cohesion and are susceptible to erosion, both through eolian processes and overland flow (Arnalds, 2004). Of these processes, wind transport of soils is widespread and significant in Iceland, as displayed by characteristic 'rofabard' features (Arnalds, 2000). Because of this wind transport of soils, Iceland is a significant source of dust on a

global scale (Arnalds et al. 2016). A comparison of modern winter wind speed and lake sediment shows good correspondence during the instrumental record in northwest Iceland (Geirsdóttir et al., 2009). We do not discount that soil erosion happens due to overland flow, but conclude that wind is the dominant driver. This part of the manuscript has been expanded as suggested.

Arnalds, O., 2004: Volcanic soils of Iceland. Catena 56:3–20.

Arnalds O., 2000. The Icelandic "Rofabard" soil erosion features. Earth Surf Process Landforms 25:17–28.

Arnalds et al., 2016. The Icelandic volcanic aeolian environment: Processes and impacts – A review. Aeolian Research 20,176-195.

Comment 3 - A few interesting questions remained outside the scope of the paper, although they might be interesting for the reader. It would be helpful to compare the reconstruction provided in this paper with those based on pollen analyses. Are they coherent?

Response: As pointed out by the reviewer, a comparison between the reconstructions provided in this paper with those based on pollen analyses is beyond the scope of this paper. However, we agree that it would only strengthen our results to point out the coherency between published pollen records from Iceland and our results. Most, if not all, pollen studies in Iceland do show similar changes occurring between 4.5 and 4.0 ka (e.g. Hallsdóttir, 1995). The most recent pollen study by Eddudóttir et al. (2016, 2017) conforms to the temperature decline seen in our lake records. More specifically, the authors show retreating woodland from 6000 to 4000 cal yr BP and further development from woodland to dwarf shrub heath around and after the Hekla 4 eruption when Betula nana and Empetrum nigrum pollen reappear in their record, suggesting a continued decrease in summer temperature. These authors also point out that the vegetation (and pollen) are susceptible to tephra fallout and abrasion, - the pollen records are not fully independent from the temperature forcing after both H4 and

[Figure]

H3. We have added a sentence emphasizing this in the current manuscript.

Hallsdóttir, M., 1995. On the pre-settlement history of Icelandic vegetation. Búvísindi Icel. Agr.Sci. 9, 1995: 17-29.

Eddudóttir et al., 2016. Climate change and human impact in a sensitive ecosystem: the Holocene environment of the Northwest Icelandic highland margin. Boreas 45:715-728.

Eddudóttir et al. 2017. Effects of the Hekla 4 tephra on vegetation in Northwest Iceland. Veget Hist Archaeobot.

Comment 4 - The dates of moraines are not mentioned. Do they agree with the sediment records?

Response: Our record on the onset of Neoglaciation is mostly based on lake sediments and glacier modeling from large ice caps, and not moraines, - and the discussion on the 4.2 ka event is mainly based on records indicating temperature decline or perturbations. Langjökull's efficient delivery of glacial sediments into Hvítárvatn dominates sediment accumulation in the lake, so any perturbations to the lake system, such as large fluctuations of the ice cap margin, result in changed sediment delivery. Lake sediment records also benefit from secure dating techniques, particularly when varved. Studies of glacial fluctuations during the mid- to late Holocene are relatively few in Iceland and identified moraines have been difficult to date accurately. Furthermore, these moraines have so far been outboard of small, surging mountain glaciers, which do not fully respond to climate changes. Stötter et al (1999) reported glacier advances in North Iceland ca. 5.4 ka, 4.7 ka and 3.4-3.5 ka and Kirkbride and Dugmore (2006) identified groups of moraines in the highlands of Iceland dated to ca. 5.7-5.2 ka, 3.8-3.2 ka. Both these records support our conclusion in the current paper that Neoglaciation in Iceland commenced between 5.5 and 5.0 ka.

Stötter, J., Wastl, M., Caseldine, C., Haaberle, T., 1999. Holocene palaeoclimatic re-

constructions in Northern Iceland: approaches and results. Quaternary Science Reviews 18, 457–474.

Kirkbride, M.P., Dugmore, A.J., 2006. Responses of mountain lee caps in central Iceland to Holocene climate change. Quaternary Science Reviews 25, 1692–1707.

Comment 5 - The modern warming is not manifested in the records (see fig. 5c). What is going on nowadays in this region? I think a few words on the current climate trends (including seasonality) in the region are necessary.

Response: We feel that discussing modern climate trends is beyond the scope of this paper. As the title states, this paper focuses on the inception of Neoglaciation and the 4.2 ka event.

Comment 6 - The Medieval Climatic Anomaly is not mentioned and discussed here. What is your opinion on this? All lake records indicate a strong decline in temperature _1.5 ka. Can you suggest an explanation for the absence of the MCA in this region unlike some other areas in the North Atlantic?

Response: As pointed out previously, the scope of this paper is the onset of Neoglaciation and the relation to the 4.2 ka event. We have previously published several papers on the last 2 ka (e.g., Geirsdóttir et al., 2009; Larsen et al., 2011; Miller et al., 2012; Harning et al., 2016), which reflect both the strong decline in temperature after 1.5 ka as well as the Medieval Climate Anomaly (MCA) and the Little Ice Age. Those papers clearly show that the MCA is not absent in our records especially when compared to the last 1.5 ka. But if we are to compare to the last 8 ka, the smaller magnitude changes of the MCA-LIA anomalies become obscured in the large-scale changes of the first-order Holocene temperature decline.

Geirsdóttir Á, Miller GH, Thordarson T, Ólafsdóttir KB. 2009b. A 2000 year record of climate variations reconstructed from Haukadalsvatn, West Iceland. J Paleolimnol 41:95–115.

[Figure]

Larsen DJ, Miller GH, Geirsdóttir Á, Thordarson T. 2011. A 3000-year varved record of glacier activity and climate change from the proglacial lake Hvítárvatn, Iceland. Quat Sci Rev 30:2715–2731.

Miller GH, Geirsdóttir Á, Zhong Y, Larsen DJ, Otto-Bliesner BL, Holland MM, Bailey D a., Refsnider K a., Lehman SJ, Southon JR, Anderson C, Björnsson H, Thordarson T. 2012. Abrupt onset of the Little Ice Age triggered by volcanism and sustained by sea-ice/ocean feedbacks. Geophys Res Lett 39:L02708

Harning, D.J., Geirsdóttir, Á., Miller, G.H., Anderson, L.,2016a. Episodic expansion of Drangaj€, Vestfir›ir, Iceland over the last 3 ka culminating in its maximum dimension during the Little Ice Age. Quat. Sci. Rev. 152, 118-131.

Comment 7 - I am not sure why the authors limit themselves by the Neoglacial time having the complete Holocene records? I would suggest to reconsider the title. They state that the 4.2 ka event is undistinguishable in the period 4.0 - 4.5 ka, so why it should be the focus and mentioned even in the title?

Response: This paper is being submitted to a special issue of CoP that is focusing on the 4.2 ka event. Although we do agree that our complete Holocene record deserves a special paper, we would like to see such a paper discuss evenly all the different perturbations that have taken place during the last 10 ka – such a paper would be more suited as a review paper and needs to be much longer with different focus than the one for this special issue.

A primary conclusion of this paper focuses on the prominent step towards cooling at 4.5-4.0 ka being statistically indistinguishable from the 4.2 ka event and coincident with Hekla 4, one of the largest explosive eruptions of the Holocene in Iceland. We state that we do see abrupt temperature decline between 4.5 and 4.0 ka, a period that also includes the timing of the Hekla 4 eruption. However, because of the uncertainties in the 14C dates we cannot say whether that temperature decline is due to the volcanic eruption or preceded the eruption, and connected to the more widespread changes

associated with the 4.2 ka event.

Comment 7 - Technical corrections ‹ËŸa 10 line 4 The current ELA pattern reflects reflect the patterns and temperature and precipitation across Iceland – please edit

Response: Thank you – this typo has been corrected

————————————————

[Figure]

Clim. Past Discuss.,
https://doi.org/10.5194/cp-2018-130-AC2, 2018

[Figure]

Reviewer's Summary comments: This paper focuses on the paleolimnological evidence for episodic cooling on Iceland ca. 4,200 yrs ago, a time associated with numerous other climate changes around the world, and particularly in the North Atlantic. The manuscript synthesizes previously published data from the same group of authors using factor analysis to examine commonality in the signals among the various records, across lake types and proxy types. The paper is an extension of a previous analysis by Geirsdottir and others, but with additional sediment records included. The study is inherently valuable because it brings together a large body of work produced by a single research group during the past decade or so, the records are of high quality, and the age models are good. I think the results will be eventually suitable for publication

in CoP, but I do have some questions and comments that I would like the authors to address. Many of my comments seek clarification of points that are being made in the discussion that I think are too vague and more transparency concerning the basis for determining ages of glacial inception. In a bigger sense, though, I think the manuscript would be improved if it had a more thorough discussion of all the stepwise cooling events that are revealed by the sediment record synthesis, not only the cooling event around 4.2ka.

Response: We thank the reviewer for the thorough comments and suggestions that help clarify and improve the paper. Firstly, we kindly remind the reviewer that this paper is being submitted to a special issue of CoP that is focusing on the 4.2 ka event - see below:

A special issue on the 4. 2ka event: The ∼4.2–3.9 ka BP abrupt aridification and cooling event (Zanchetta et al., 2015; Weiss, 2016) is recognized in many locations across the globe, but its causes, precise timing, characteristics and quantification remain enigmatic. A 3-day international workshop on this topic was held at the Dipartimento di Scienze della Terra (Università di Pisa, Italy), 10–12 January 2018, and attended by ∼60 people. This special issue will include individual papers presented at the meeting and regional syntheses subsequently developed by those in attendance.

Although we do agree that our complete Holocene record deserves a special paper, we would like to see such a paper equally discuss all the different perturbations that have taken place during the last 10 ka. Such a paper would be more suited as a review paper and requires a different focus than the one for this special issue, which requires some focus on the 4.2 ka event.

Below we address all of this reviewer's comments and concerns to the best of our ability. Corresponding revisions will be made to the manuscript.

Reviewer comments: 1. The title of the paper includes the phrase "onset of neoglaciation," and I appreciate that the authors include a discussion of the origin of this term, but

when do they believe this onset began in Iceland? It seems this should be a fundamental conclusion of the paper if it is so prominent in the title. The discussion leads one to understand that neoglacial inception was catchment-specific across Iceland, and this is to be expected. Yet, this seems at odds with the conclusion that the neoglacial began at _5ka (which really comes from Larsen et al., 2012). I don't follow the logic used to determine this timing. Why are the BSi records shown in Fig 6 and 7 used to justify this timing for glacial inception? Do the blue bars on this figure denote onset of minreogenic input to the lakes from glacial erosion? I have either missed it in the discussion, or it could use further explanation. As it stands, I do not find that a neoglacial onset of 5.5ka can be concluded from the records shown. What provides the basis for this?

Response: We refer to page 9, line 33 and page 10, line 1 to 3 and again page 10 lines 7-19. Regarding the onset of Neoglaciation/first glacier inception we do point out that Icelandic glaciers are most sensitive to temperature and that we expect the timing of glacier inception to be controlled by the rate at which Holocene temperature declines. BSi currently reflects best the temperature change in our records. We also say on page 9, line 33: Following Porter's (2000) definition, the onset of Neoglaciation in Iceland based on our lake records occurred before 5.0 ka for Langjökull, although the initial growth of Drangajökull occurred much later, $\sim$2.3 ka. This indicates that the spatio-temporal nucleation of glaciers in Iceland was indeed asynchronous and likely reflects the relation between the regional ELA and topography, but does not change our interpretation of when the Neoglacial onset began (i.e. $\sim$5.5 ka). The nature of the topography (i.e., the hypsometry) controls how quickly the glacier will expand after the ELA intersects the topography. This is an important result as defining the onset of the Neoglaciation in Iceland would depend on which ice cap one studied. If the study solely relied on the datasets from Drangajökull, the conclusions would be very different compared to the conclusions drawn when two additional (and high elevation) ice caps are included.

To clarify this and to improve the flow of the paper we have rearranged the sections

within the Discussion so that we now discuss the 4.2 ka event first with focus on the BSi record as a spring/summer temperature indicator. Here we also extend the description of the sedimentological parameters that do support glacier activity in HVT's catchment such as the MS record and the sediment accumulation rate. We then discuss the demise and growth of the Icelandic glaciers during the Holocene, and finally we discuss the onset of Neoglaciation.

The conclusion is as pointed out in the manuscript that the onset of Neoglaciation in Iceland did indeed occur with the inception of Langjökull although other glaciers didn't start to nucleate until around 4.5 ka or later, which emphasizes the importance of the 4.5-4.0 ka temperature decline.

2. Section 5.2 is called: The onset of neoglaciation. Section 5.3 is called: The onset of neoglaciation in the circum North Atlantic and the 4.2 ka event. I think the authors intended to have a greater discussion of the North Atlantic patterns of / evidence of neoglacial inception in different regions in section 5.3, but this never materializes. It probably should, because there is ample evidence from many parts of the North Atlantic. This discussion should either be expanded or the section title changed to reflect the content. The discussion of neoglaciation in the circum North Atlantic seems too brief. It basically stops at noting that records appear to be driven by monotonic insolation forcing. This is a shame, because there are other studies that previously showed stepwise cooling in the Holocene (including at 4.2ka) and that provide support for the authors' interpretations. These should be discussed/cited.

Response: We acknowledge this comment and refer to our response to comment 1 above. We have now changed the title of the section. The focus here is on the stepwise temperature decline as seen in the BSi record, the 4.2 ka event and relation of the Neoglaciation of Iceland. We cite similar studies from the circum-North Atlantic area as supportive research for similar ELA lowering, but have decided that it is not necessary to review studies on Holocene stepwise climate changes in the circum-North Atlantic area in depth in this paper.

[Figure]

3. I understand that this is for a special issue of CoP concerning climate changes that took place around 4.2ka and that this motivation has steered the direction of the discussion of the datasets being synthesized in this manuscript. However, it seems like a missed opportunity to focus only on this single event. Doing so implies to the reader that the changes in the records observed at 4.2ka are somehow bigger, more abrupt, or different in some way from the many other abrupt changes seen in these records. What about the 6.5ka, 5.5ka, 3.0ka, and 1.5ka Events? These all stand out as equally important and noteworthy in the 7-lake all proxies record. The fact that the high latitude North Atlantic cooled through the mid-late Holocene in a stepwise manner is very important. I think a section that discusses the other abrupt cooling steps and their relationships to potential forcing factors, oceanographic changes, etc, as has been done for the changes between 4-4.5ka, would be very valuable.

Response: We refer the reviewer to Geirsdóttir et al. (2013), which discusses the overall stepwise changes during the Holocene. Because we point out in the text (and Figure 5) the same stepwise pattern is very apparent in the all lake composites, we do not feel like that it is necessary to repeat the Geirsdóttir et al. (2013) conclusions. Instead, we place the focus here on the appearance of the 4.2 ka temperature decline and its relation to the nucleation of current glaciers in Iceland and the global 4.2 ka event. Our record from HVT (both physical and biological) shows the first major and apparent temperature change between 5.5 and 5.0 ka, which is further supported by glacier modeling indicating glacier inception at this time. Hence, 5.5 ka is an important point in time in regard to the focus of the paper. All the other lakes show the first synchronized/correlative big change between 4.5- 4.0 ka, which allows us to tease apart its potential relationship to the global 4.2 ka event. Hence 4.2 ka is another important step to focus on in this paper, especially given the focus of the CoP special issue.

We have added a paragraph at the end of the Introduction as followed to clarify the aim of the paper: In order to understand the non-linear pattern and stepped changes in

[Figure]

Iceland after the HTM (Geirsdóttir et al., 2013), and how regional temperatures evolved in terms of timing, magnitude and glacier inception, we focus specifically on the climate steps between 6.0 and 3.0 ka. This time interval includes the 4.2 ka aridification and cooling event recognized at many global locations across latitudes and longitudes. We place the 4.2 ka event in the context of our Icelandic Holocene climate reconstruction and knowledge of large Icelandic volcanic eruptions as a way of judging if it is indeed a major climate event.

4. There is an implication running through his manuscript that the climate perturbations seen in these records are due to climate cooling episodes caused by volcanism. This is possibly true. But I think I think this hypothesis should be given it's own discussion rather than being inserted here and there throughout the manuscript. For example, the first line in the method section presupposes that volcanoes are the primary climate forcing responsible for the signals in the proxies – I don't really think this is an appropriate place in the manuscript to insert this concept. Something that strikes me as particularly confusing is that one of the conclusions of the paper is that the Hekla4 eruption, although coincident with the 4.2 event, could not be responsible for the climate cooling because of low SO2 in the eruption. Yet, there is wording throughout the manuscript that leads a reader to believe that volcanism was responsible for climate cooling events observed in the record. I think that a section that specifically focuses on he role of volcanoes on climate and on landscape dynamics would be very useful.

Response: We have amended the text to reflect two different effects of volcanism where appropriate (i.e., tephra deposition and aerosol production), which were likely not clear to the reviewer beforehand. One effect is the local tephra deposition that results in vegetation destruction and soil erosion. Because this is manifested as increased C/N in the lake sediment records, which we typically interpret as cool and windier winters, it potentially obscures the temperature records we are aiming for. The other effect is emission of aerosols and gasses from volcanic eruptions, which would impart a climatic effect. We have changed our wording throughout the text to reflect

our intending meaning of the two possibilities.

To prevent any confusion, we have now removed the first sentence in the Method chapter, "that volcanoes are the primary climate forcing responsible for the signals in the proxies." We have also extended our discussion on the impact of explosive volcanism and tephra fallout on the catchments and lake proxies in section 2 and 3.1, but do not feel an entirely separate section is needed at this point.

5. The authors interpret C/N and TOC as both directly related to catchment erosion (pg5 lines 1,2). This may be the case. If so, it can be easily tested by comparing the two data sets. Do TOC and C/N carry the same signal within each lake? This would support the interpretation. How do they correlate within each record? I ask because it seems these signals could be much more complicated. It isn't clear to me how colder conditions lead to greater soil erosion. Is it because there is less vegetation during cold times? Or is it because there is greater glacial erosion? That shouldn't matter in the non-glaciated catchments. Moreover, when BSi decreases shouldn't the in-lake organic productivity also go down, leading to higher C/N even in the absence of changes in terrestrial input? When BSi increases due to increased productivity, won't TOC go down due to dilution of the sediments by BSi, even in the absence of changes in soil erosion? It seems like mass accumulation rates are needed to consider these proxies independently, particularly in the glaciated catchments where changes in glacially derived material is likely the primary control on all of these other measured proxies (as %).

Response We have now made corresponding revision to the manuscript where we have added a brief discussion of the reasoning for using these proxies and their interpretation. For a detailed explanation, we direct the reader to the 2013 paper, which indeed assesses the relationship between TOC and C/N. To specifically address both reviewers' question about TOC, we suggest that TOC in the sediment is a product of both production and transport terms (among other factors). The production term increases during warm periods due to increased plant growth, but transport from land is

reduced as vegetation growth stabilizes slopes and carbon accumulates (and remains sequestered) in soils. During cold periods, even though the autochthonous production term is minimized, catchment vegetation is reduced allowing increased transport of previously accumulated organic matter from eroding soils resulting in a large influx of terrestrial OC to the lake sediment. This more than compensates for any decrease in productivity due to shorter growing seasons and leads to a net increase of lake sediment TOC during cold periods. The soils of Iceland lack cohesion and are susceptible to erosion, both through eolian processes and overland flow (Arnalds, 2004). Of these processes, wind transport of soils is widespread and significant in Iceland, as displayed by characteristic 'rofabard' features (Arnalds, 2000). A comparison of modern winter wind speed and lake sediment shows good correspondence during the instrumental record in northwest Iceland (Geirsdóttir et al., 2009). We do not discount that soil erosion happens due to overland flow or glacier erosion, but conclude that wind is the dominant driver, particularly since most of our lakes are non-glacial. This part of the manuscript has been expanded as suggested.

6. Ideally, a synthesis of various records would include error bounds that propagate the uncertainty in the age models of each sediment core. I know the age models are quite good in these records, due to the abundant tephra layers, but correlating the records of lakes to within a couple hundred years is still quite challenging. The correlation uncertainties change as a function of distance from age control points, and the authors have already calculated the age uncertainties for each sediment record using BACON. These should really now be used to propagate these uncertainties into the "all proxy" composites. The abrupt changes are very evident and I do not doubt them and I think they will remain a robust feature, but this uncertainty analysis would be useful.

Response: we note the comment and will in future papers work to propagate uncertainties more rigorously. However, The tephra-based chronologies for each lake sediment sequence and correlation between lakes based on the same tephra layers together with synchronization of paleomagnetic secular variation between four of the lakes (HVT,

[Figure]

HAK, ARN, TORF) and a very well dated marine sediment core off the coast of northern Iceland, provide robust chronologic control and minimal age uncertainties over the Holocene (e.g., Jóhannsdóttir, 2007; Stoner et al. 2007; Ólafsdóttir et al 2013; Harning et al., 2018a). The age model for each lake was constructed by fitting control points with a smoothed spline using the CLAM code (Blaauw, 2010). Analyseries software (Paillard et al., 1996) was used to resample each proxy time series to the same 20-year increments before making the composites.

7. One thing that is unclear to me in the manuscript at times is what is meant by "volcanic impact" on a catchment. It seems as though this is sometimes referring to the indirect impact of volcanism via its impact on climate, and at other times is referring to the actual physical impact on a specific catchment stemming from local volcanism. One example of this is line 22 on page 10, where the authors refer to the greater impact on HVT and ARN (relative to the other lakes) from catchment-specific processes, including volcanism. I can't tell whether this is implying that volcanism leads to changes in catchment erosion independent of climate that then obscures the climate records, or if the point is that these catchments respond differently to external climate forcing because they are more continental than the coastal sites and a given volcanic eruption impacts them more. I believe that this could use clarification. A dedicated section about the impacts of volcanism on lake records would probably be very useful in clarifying the authors' meaning.

Response: See response to comment 4.

8. Page 7, line 26: "Sedimentological analyses of HVT: : :toward cooler conditions: : :" Fig. 3a is referenced here, but there is no BSi record on that figure and the "all proxy record" from HVT shown on Fig 3 actually shows a first cooling step at 6.5ka, not 5.5ka. Is there some other information being used to assign the 5.5ka step as the neoglacial inception? Perhaps mag susceptibility, grain size, or minerogenic content? Especially because "neoglacial onset" is in the title of the paper, it seems that being as clear as possible about the underlying evidence is important. Also, is this change in

[Figure]

BSi actually diminished biological productivity, or driven more by the dilution impact of renewed input of glacially derived sediments to the lake? Seems the latter is a better indication of glacier inception.

Response: see our response to comments 1 and 2 - We have now changed the figure reference to Fig. 3a,b – as described in the text the all composites includes both temperature change and catchment reactions whereas the BSi (Fig. 3b) is more indicative of the temperature decline. . .

9. Page 8, line 2-4: "The impact of the tephra on the landscape in either case is unambiguous: : :" What is the "unambiguous impact" of the tephra on the landscape? There is an implication here that the tephra somehow impacts the catchment response to climate forcing, or maybe confounds the proxies in the lakes of those catchments such that they do not represent climate when there is tephra in the catchment – but exactly how this works and the impact on the proxy interpretations is never discussed. I'd like a more detailed discussion of these impacts.

Response: see response to comments 4 and the amended text.

10. Line 24 page 11: The final words here are "supporting our conclusions," but it is not clear what is meant. I think the authors are saying that other studies that have either documented or speculated about abrupt changes in ocean currents during the past 2ka support their interpretation that the lake sediments document abrupt cooling events. This isn't really true, first of all; but secondly the wording is too vague and I don't understand what exactly is being linked between these cited papers that have interpretations about the internal feedbacks of the North Atlantic and the conclusions of this paper, which at this point in the paper have not yet been reached, or at least are still a bit vague.

Response: The text has been revised as follows: Although the gradual decline in summer insolation progressively lowered the ELA, the significant stepwise trend in the Icelandic records suggests that strong local to regional feedbacks modulated the primary

insolation forcing. The rate of cryosphere expansion at 4.5-4.0 ka and particularly after 1.5 ka suggests contemporaneous shifts in the northern North Atlantic region. Such episodic ice expansion cannot be explained by the summer insolation forcing alone and requires additional forcing or changes in North Atlantic circulation.. Variations in the strength of the thermohaline circulation, weakening of the northward heat transport of the AMOC and/or increasing influence of the Arctic waters influence all these locations. Changes in the strength of AMOC and/or the subpolar gyre and changes in the Arctic sea ice extent with the associated meridional heat transport into the Arctic have been related to past cooling events, particularly during the last 2 ka (Trouet et al., 2009, 2012; Lehner et al., 2013; Moreno-Chamarro et al., 2017; Zhong et al., 2018).

11. Conclusions: There are 6 conclusions of the paper provided as a bulleted list. Some of these are not actually conclusions that can be drawn from the results presented here, and are better described as discussion points than conclusions. Some are not necessarily supported by the data presented. I would ask the authors to be more specific about their conclusions, remove those that aren't really conclusions from this study, and to provide them as a narrative rather than a list, so they can explain/summarize. 1. ELA intercepted Langjokull at 5ka. There isn't evidence for this presented in this paper. 2. This conclusion is saying that the Holocene cooling on Iceland happened in a stepwise manner, which I think is a reasonable conclusion. However, based on the records presented, the first cooling event happened well before 5ka. 3. Stepwise cooling requires internal feedbacks, which possibly involve ocean dynamics. Reasonable conclusion. 4. I think this conclusion is that Hekla 4 eruption did not cause the cooling associated with the 4.2 event, even though they are contemporaneous. Does this conclusion really stem from the results presented in this study? 5. This is about sea ice expansion during neoglaciation, but this isn't a conclusion from this Iceland lake synthesis. 6. Ocean circulation influenced climate on Iceland. This can be a conclusion based on the comparison of the Iceland records with some marine records, but the conclusion should be much more specific than this. Clearly, ocean dynamics impact Iceland's temperature – isn't there more than can be concluded about

how and when?

Response: We have now edited the conclusions to reflect what we are deriving from the dataset presented.

Tables: Data tables are incomplete. Table 1 – Can easily delineate and measure the catchment area of the two lakes that aren't included using readily available maps. Table 1 – title says core description, but the table contains only lake descriptions.

Response: Acknowledged. Has been corrected.

12. There are some typos throughout that the author's should look out for. But one in particular that spellcheck won't pick up on is in the Abstract, where is says decent instead of descent.

Response: Acknowledged. Has been corrected.
* * *
[Figure]

**The onset of Neoglaciation in Iceland and the 4.2 ka event**

Áslaug Geirsdóttir[1], Gifford H. Miller[1,2], John T. Andrews[2], David J. Harning[1,2], Leif S. Anderson[1,3], Christopher Florian[2], Darren J. Larsen[4], Thor Thordarson[1]

5  [1]Faculty of Earth Science, University of Iceland
[2]INSTAAR/Department of Geological Sciences, University of Colorado Boulder
[3]Department of Earth Sciences, Simon Fraser University, CANADA
[4]Department of Geology, Occidental College, Los Angeles, CA

10  *Correspondence to*: Áslaug Geirsdóttir (age@hi.is)

**Abstract.** Strong similarities in Holocene climate reconstructions derived from multiple proxies (BSi, TOC, $\delta^{13}$C, C/N, MS, $\delta^{15}$N) preserved in sediments from both glacial and non-glacial lakes across Iceland indicate a relatively warm early-to-mid

15  Holocene from 10 to 6 ka, overprinted with cold excursions presumably related to meltwater impact on North Atlantic circulation until 7.9 ka. Sediment in lakes from glacial catchments indicates their catchments were ice-free during this interval. Statistical treatment of the high-resolution multiproxy paleoclimate lake records shows that despite great variability in catchment characteristics, the sediment records document more or less synchronous abrupt, cold departures as opposed to the smoothly decreasing trend in Northern Hemisphere summer insolation. Although all lake records document a decline in

20  summer temperature through the Holocene consistent with the regular decline in summer insolation, the onset of significant summer cooling, occurs ~5 ka in high-elevation interior sites, but is variably later in sites closer to the coast, suggesting proximity to the sea may modulate the impact from decreasing summer insolation. The timing of glacier inception during the mid-Holocene is determined by the descent of the Equilibrium Line Altitude (ELA), which is dominated by the evolution of summer temperature as summer insolation declined as well as changes in sea surface temperature for coastal glacial

25  systems particularly in coastal settings. The glacial response to the ELA decline is also highly dependent on the local topography. The initial ~5 ka nucleation of Langjökull in the highlands of Iceland defines the onset of Neoglaciation in Iceland. Subsequently, a stepwise expansion of both Langjökull and northeast Vatnajökull occurred between 4.5 and 4.0 ka, with a second abrupt expansion ~3 ka. Due to its coastal setting and lower topographic threshold, the initial appearance of Drangajökull in the NW of Iceland was delayed until ~2.3 ka. All lake records reflect abrupt summer temperature and

30  catchment disturbance at ~4.5 ka, statistically indistinguishable from the global ~4.2 ka event, and a second widespread abrupt disturbance at 3.0 ka. Both are intervals characterized by large explosive volcanism and tephra distribution in Iceland resulting in intensified local soil erosion, but unlikely to affect regional climate. The most widespread increase in glacier advance, landscape instability, and soil erosion occurred shortly after 2 ka, likely due to a complex combination of increased impact from volcanic tephra deposition, cooling climate, and increased sea ice off the coast of Iceland. All lake records

Áslaug Geirsdóttir 12/18/2018 2:11 PM

Áslaug Geirsdóttir 12/18/2018 2:12 PM

Áslaug Geirsdóttir 12/18/2018 2:14 PM

Áslaug Geirsdóttir 12/18/2018 2:15 PM

Áslaug Geirsdóttir 12/18/2018 2:15 PM

Áslaug Geirsdóttir 12/18/2018 2:16 PM

Áslaug Geirsdóttir 12/18/2018 2:16 PM

Áslaug Geirsdóttir 12/18/2018 2:18 PM

Áslaug Geirsdóttir 12/18/2018 2:17 PM

Áslaug Geirsdóttir 12/18/2018 2:19 PM

Áslaug Geirsdóttir 12/18/2018 2:19 PM

Áslaug Geirsdóttir 12/18/2018 2:20 PM

indicate a strong decline in temperature and increase in glacier expansion ~1.5 ka, which culminated during the Little Ice Age (1250-1850 CE) when all ice caps reached their maximum dimensions.

**1 Introduction**

5    A growing number of proxy-based climate reconstructions provide essential information for understanding the patterns, and mechanisms behind millennial-to-centennial scale climate variability during the Holocene epoch. Most Northern Hemisphere reconstructions illustrate warmer-than-present conditions during the Holocene Thermal Maximum (HTM, 11-6 ka) but display spatiotemporal heterogeneity. For example, the HTM in the northern North Atlantic may have been delayed by up to 3000 years compared to the western Arctic (Kaufman et al., 2004). This asynchrony has been attributed to the lingering

10    effects of the residual Laurentide Ice Sheet (LIS) and the Greenland Ice Sheet (GIS), which affected surface energy balances and impacted ocean circulation due to variable meltwater fluxes (Barber et al., 1999; Renssen et al., 2009, 2010; Blaschek and Renssen, 2013; Marcott et al., 2013; Sejrup et al., 2016). Although the primary forcing of the Northern Hemisphere's subsequent cooling was the spatially uniform decline in summer insolation (Berger and Loutre, 1991), orbital forcing also led to nonuniform changes within the climate system (e.g. Wanner et al., 2008). On decadal to multi-centennial timescales,

15    Holocene climate variability is linked to forcing factors such as solar activity and large tropical volcanic eruptions, in addition to coupled modes of internal variability such as the North Atlantic Oscillation (NAO), and changes in the Atlantic Meridional Overturning Circulation (AMOC) and its capacity to transport heat in the North Atlantic. Feedbacks between ocean, atmosphere, sea ice and vegetation changes also complicate the global response to primary insolation forcing, and result in non-linear climate variability (Trouet et al., 2009; Miller et al., 2012; Lehner et al., 2013; Moreno-Chamarro et al.,

20    2017).

A recent synthesis showed that a general trend of mid Holocene glacier growth in the Northern Hemisphere was a response to declining summer temperatures driven by the orbitally-controlled reduction of summer insolation (Solomina et al., 2015). However, spatially divided compilations based on the Arctic Holocene Transitions (AHT) database (Sundquist et al., 2014) indicate heterogeneous responses to the cooling between Arctic regions (Briner et al., 2016; Kaufman et al., 2016;

25    Sejrup et al. 2016). The response of northern North Atlantic region is considered to be particularly complex. Records from this region show a strongly uniform pattern of sea surface and terrestrial summer temperature responses to changes in summer insolation, but also signatures of sub-regional differences that were apparently related to both the strong influence of the LIS's disintegration and meltwater discharge in the early Holocene and the variability in AMOC strength throughout the remainder of the Holocene (Sejrup et al., 2016).

30    Iceland lies in the northern North Atlantic, a region strongly affected by the northward heat transport of AMOC (Fig. 1), and within the periphery of the strongest contemporary warming (Overland et al., 2016). Paleoclimate research across Iceland has shown that positive summer insolation anomalies of the early Holocene resulted in an "ice-free" Iceland by ~9 ka

Áslaug Geirsdóttir 12/3/2018 1:15 PM

Áslaug Geirsdóttir 12/18/2018 2:21 PM

Áslaug Geirsdóttir 12/3/2018 1:16 PM

Áslaug Geirsdóttir 12/18/2018 2:21 PM

Áslaug Geirsdóttir 12/18/2018 2:22 PM

Áslaug Geirsdóttir 12/18/2018 2:27 PM

Áslaug Geirsdóttir 12/18/2018 2:24 PM

Áslaug Geirsdóttir 12/18/2018 2:24 PM

Áslaug Geirsdóttir 12/18/2018 2:25 PM

Áslaug Geirsdóttir 12/18/2018 2:27 PM

Áslaug Geirsdóttir 12/18/2018 2:27 PM

(Larsen et al., 2012; Striberger et al., 2012; Harning et al., 2016a). Glacier modeling experiments suggest that the observed disappearance of Langjökull in central Iceland by 9 ka required summer temperature 3°C above the late 20th Century average (Flowers et al., 2008), an estimate reinforced by simulations for Drangajökull, an ice cap in northwest Iceland (Anderson et al., 2018). Reconstructions from two high-resolution lake-sediment records in Iceland show that summertime cooling during the Holocene had commenced by ~5 ka (Geirsdóttir et al., 2013). This cooling was likely intensified by lower sea-surface temperatures around Iceland and increased export of Arctic Ocean sea ice resulting in renewed glacier nucleation in the mid Holocene (Geirsdóttir et al., 2009a; Larsen et al., 2012; Cabedo-Sanz et al., 2016). The high-resolution lacustrine records indicate that, despite the monotonic decline in summer insolation, Iceland's landscape changes and ice cap expansions were non-linear with abrupt changes occurring ~5.0, ~4.5-4.0 ka, ~3.0 ka and ~1.5 ka (Geirsdóttir et al., 2013) with Iceland's ice caps reaching maximum dimensions during the Little Ice Age (LIA, ~1250-1850 CE; Larsen et al. 2015, Harning et al., 2016b; Anderson et al., 2018). In order to understand the non-linear pattern and stepped environmental and climate changes in Iceland after the HTM, and how regional temperatures evolved in terms of timing, magnitude and glacier inception, we focus specifically on the climate steps between 6.0 and 3.0 ka. This time interval includes the 4.2 ka aridification and cooling event recognized at many global locations. We place the 4.2 ka event in the context of our Icelandic Holocene climate reconstruction and knowledge of large Icelandic volcanic eruptions as a way of judging if it is indeed a major climate event.

This paper composites six different normalized climate proxies (i.e. mean = 0 ± 1) in high-sedimentation-rate cores from seven lakes in Iceland. Here, we add five new sediment records to the original two-lake composite of Geirsdóttir et al. (2013) to more effectively evaluate whether individual site-specific proxy records reflect regional climate changes or catchment-specific processes that are less directly coupled to climate. We employ a statistical comparison of our proxy reconstructions from the seven lakes to quantitatively evaluate their between-lake correlation over time and by inference, their utility as proxies for regional climate.

**2 Study sites**

The temperature and precipitation gradients across Iceland reflect the island's position between cold sea-surface currents from the Arctic and warmer, saltier sea-surface currents from lower latitudes delivered by discrete limbs of the AMOC (Fig. 1). Consequently, climate perturbations that alter the strength, temperature, and/or latitudinal position of these currents should translate to significant changes in the mean terrestrial climate state of Iceland. As an example, sea ice imported and locally formed off the north coast of Iceland during extreme cold events provides a positive feedback on the duration and severity of short-term perturbations from volcanic eruptions (Miller et al. 2012; Sicre et al., 2013; Slawinska and Robock, 2018).

Due to Iceland's location astride the North Atlantic Ridge, frequent volcanism during the Holocene has influenced Iceland's environmental history. The basaltic bedrock generated during the island's volcanic origins is easily erodible, which results in high sedimentation rates within Icelandic lake catchments and the potential for high-resolution climate records.

Áslaug Geirsdóttir 12/4/2018 8:39 PM

Áslaug Geirsdóttir 12/4/2018 8:39 PM

Áslaug Geirsdóttir 12/3/2018 1:51 PM

Áslaug Geirsdóttir 12/3/2018 1:51 PM

Áslaug Geirsdóttir 12/18/2018 2:29 PM

Áslaug Geirsdóttir 12/18/2018 2:29 PM

Áslaug Geirsdóttir 12/3/2018 1:51 PM

Áslaug Geirsdóttir 12/18/2018 2:30 PM

Áslaug Geirsdóttir 12/18/2018 2:32 PM

Áslaug Geirsdóttir 12/18/2018 2:33 PM

Áslaug Geirsdóttir 12/18/2018 2:34 PM

The frequent production of thick tephra layers have also periodically stripped the landscape of vegetation and triggered intensified soil erosion (cf., Arnalds 2004; Geirsdóttir et al., 2009b; Larsen et al., 2011; Blair et al 2015; Eddudóttir et al., 2017). Impacts on the climate and regional temperatures from Icelandic volcanism, however, has yet to be documented. Clear evidence exists for widespread soil erosion across Iceland during the past thousand years. While the prevailing

5    hypothesis favors human settlement as the trigger for widespread erosion, a combination of natural late Holocene cooling and volcanic tephra deposition, also likely played a role (e.g., Geirsdóttir et al, 2009b).

The seven lakes studied here lie on a transect crossing the temperature and precipitation gradient from south to northwest (Table 1, Figs. 1, 2). Furthermore, they represent a range of catchment styles, including lowland/coastal lakes, highland lakes, non-glacial and glacial lakes, and span elevations between 30 and 540 m asl. The lakes are of varied sizes,

10   from 0.2 to 28.9 km$^2$, and lake depths range from 3 to 83 m (Fig. 2, Table 1). In most years, the lakes are ice covered from November to April, although early winter thaws and/or late season ice cover may occur. All lakes contain a high-resolution Holocene record with sediment thickness ranging from 2.5 m (SKR) to >20 m (HVT) and occupy glacially scoured bedrock basins. Vestra Gíslholtsvatn (VGHV), Arnarvatn Stóra (ARN), Torfdalsvatn (TORF), Haukadalsvatn (HAK) and Skorarvatn (SKR) have not received glacial meltwater since deglaciation and preserve organic rich sediments deposited throughout the

15   remainder of the Holocene. In contrast, Hvítárvatn (HVT) and Tröllkonuvatn (TRK) are glacial lakes that have received glacial meltwater when ice margins re-occupied lake catchments following early Holocene deglaciation. Shrub-heath characterizes the vegetation around most of the lakes, with low-lying plants and mosses dominating the shallow soils. The one exception is VGHV, the southernmost lake, which today is predominately surrounded by large planted hayfields. However, shrub-heath grows near bedrock outcrops on elevated areas around VGHV, whereas fen/mires prefer locations

20   where water can accumulate in lower elevation areas. Three of the lakes, VGHV, HVT and ARN, lie within the current active volcanic zone of Iceland, and therefore contain and preserve more frequent and thicker tephra layers compared to the other four lakes that all lie distal to the main volcanic zones (Fig. 1).

**3 Methods**

25   Geirsdóttir et al. (2013) composited six different normalized climate proxies in high-sedimentation rate (≥1 m ka⁻1) cores from two lakes: a glacial (HVT) and a non-glacial (HAK) lake. Despite the different catchment-specific processes that characterized each lake's catchment, the composite proxy records show remarkably similar stepped changes toward colder states after the mid-Holocene. The common signal in very different catchments indicates that the climate proxies in the sediment records reliably reflect past climate change in Iceland. Here we add five new Icelandic lake sediment records and

30   test whether we can replicate the two-lake composite record in other lakes across Iceland. First, we generate a single multi-proxy composite record for each lake by applying the same methodology as in Geirsdóttir et al. (2013). Before calculating each regional mean record, all records were standardized over the whole record/period, which permits reliable comparison of the records with each other.
* * *
Áslaug Geirsdóttir 12/18/2018 2:35 PM

Áslaug Geirsdóttir 12/18/2018 2:35 PM

Áslaug Geirsdóttir 12/18/2018 2:36 PM

**3.1 Generating composite records from physical and organic proxies**

The seven multi-proxy lake composites were generated following Geirsdóttir et al. (2013). Individual composite records were calculated from normalized values for each of six environmental proxies by subtracting the physical proxies (TOC, MS, C/N), which reflect erosional activity and cold and windy seasons within the catchment, from the organic matter proxies (BSi, $\delta^{13}$C, $\delta^{15}$N), which reflect biological productivity (mainly diatom growth) and summer warmth. As explained in Geirsdóttir et al. (2009b), TOC and C/N are grouped with the physical proxies (catchment instability) in the composites due to the flux of soil carbon into the lakes during periods of severe soil erosion occurring during cold, dry and windy seasons. The TOC in the sediment is primarily a product of both production and transport. The production term is increased during warm periods due to increased plant growth, but transport is minimized as organic material in the catchment accumulates and remains sequestered in soils. During cold periods, even though the production term is minimized, the transport of previously accumulated organic matter from eroding soils contributes a large influx of OC to the lake sediment. This more than compensates for any decrease in productivity due to shorter growing seasons and leads to a net increase of sediment TOC during cold periods. The soils of Iceland lack cohesion and are susceptible to erosion, both through aeolian processes and overland flow (Arnalds, 2004). Of these processes, wind transport of soils is widespread and significant in Iceland, as displayed by characteristic 'rofabarð' features (Arnalds, 2000). A comparison of modern winter wind speed and lake sediment shows good correspondence during the instrumental record in northwest Iceland (Geirsdóttir et al., 2009). We do not discount that soil erosion happens due to overland flow, but conclude that wind is the dominant driver.

All proxies are given equal weight in each lake's multi-proxy composite (Geirsdóttir et al., 2013). Analyseries software (Paillard et al., 1996) was used to resample each proxy time series to the same 20-year increments. The 20-year interval reflects the minimum resolution within the least resolved dataset used in the analysis (i.e., the BSi record). The C/N ratio reflects the ratio of terrestrial- versus aquatic-derived carbon in the lake sediments, and hence the degree of erosion in the lake catchments. The individual proxy data used here have been published previously, and corresponding methods, geochronology, and interpretations are described in the relevant original publications (Table 1).

**3.2 Correlating lake sediment cores**

The abundant tephra layers archived in Icelandic lake sediments generally hold diagnostic geochemical fingerprints, which allow them to serve as key chronostratigraphic markers. These tephra horizons have been dated using historical information and radiocarbon measurements of soil, lacustrine and marine archives. The tephra-based chronologies for each lake sediment sequence, together with synchronization of paleomagnetic secular variation between four of the lakes (HVT, HAK, ARN, TORF) and a well-dated marine sediment core off the coast of northern Iceland, provide robust chronologic control and minimal age uncertainties (<100 years) over the Holocene (e.g., Jóhannsdóttir, 2007; Stoner et al. 2007; Larsen et al., 2012; Ólafsdóttir et al 2013; Harning et al., 2018a). These detailed chronologies thus allow for the development of regional syntheses of climate history through the direct comparison of the records from nearby lakes (Geirsdóttir et al., 2013). The

age model for each lake was constructed by fitting control points with a smoothed spline using the CLAM code (Blaauw, 2010) before resampling to the same 20-year increment. All ages in the text are reported as calendar years prior to 2000 CE (b2k).

5  **3.3 Statistical analyses**

Our primary goal when reconstructing Holocene climate evolution is to test whether changes on decade-to-century scales are regionally coherent (e.g., Geirsdóttir et al., 2013). This effort requires continuous records from different catchments where consideration is given to the large number of highly resolved climate proxies derived from each of the seven lake sediment records. We are primarily interested in the agreement or otherwise of the trends in the data. In order to compare both within-

10  lake and between-lake variables, we normalized the resampled data (n = 3478) so that: 1) all variables had equal weight (mean = $0 \pm 1$), and 2) differences between lakes, possibly due to climatic gradients, are clarified. In order to evaluate the linkages between the lake proxies we applied R-Mode factor analysis to the 6 x 3478 data matrix (Davis, 1986; Aabel, 2016) to evaluate the proxy records as indices of regional climate change, rather than site-specific environmental conditions that may be decoupled from regional climatic parameters. Factor analysis extracts the dominant signal (1st PCA), and the

15  explained variance plus each lakes factor scores gives a measure of how the different sites are associated to this main signal, hence allowing us to evaluate the importance of an individual proxy as a climate recorder.

**4 Results and interpretation**

**4.1 Proxy measurements and the composite Holocene records**

20  The multi-proxy composites for each lake reduce proxy-specific signals within each lake, while amplifying those signals that are recorded by most or all proxies (Fig. 3a). By isolating each individual lake composite record, the signal of catchment specific processes within each lake record is preserved, which validates our comparison between different catchments to test for a regional, Iceland-wide signal.

25  **4.2 Results from Factor analyses**

We employ the statistical comparison between our proxy-based reconstructions from the lakes in order to quantitatively evaluate their correlation over time and by inference, their utility as proxies for regional climate. The results (Table 2) indicate that the 1st factor explains 56.9% of the variance and has high loadings (associations) with 5 of the proxies. MS and $\delta^{15}N$ are positively loaded on this axis whereas TC, BSi, and $\delta^{13}C$ have strong negative loadings. Communalities, a measure

30  of shared information (Davis, 1986; Aabel, 2016), is high for all 6 proxies with $\delta^{15}N$ having the highest unique (noise) rating. An additional 19.5% of the variance is explained by the 2nd factor, which is largely a measure of the C/N ratio. Varimax factor analysis (Table 2) does not result in any significant changes in the rankings or sign of the proxies, except for BSi, which now has the strongest loading on factor 1. The first two factors explain 76.4% of the data indicating that most of the variance in the dataset may be explained by these factors and that the seven lake sediment records are showing a similar

signal.

A plot of the factor scores (Fig. 4), labelled for each of the 7 lakes, indicates a clear environmental gradient along the 1st factor axis and a less distinct, but still clear, division on the 2nd factor axis, for example between HVT and HAK (Fig. 4). These distinct clusters reflect the combined scaling of each 3478 measurements per proxy.

5    The similarities in timing and direction in variations of climate proxies preserved in all seven composites, our robust age models, and the communalities (Table 2), enable us to consider all seven composite lake records and generate a single time series (Fig. 5a). Compared to our previous composite (Geirsdóttir et al., 2013), the 7-lake composite is near identical indicating that all lakes experienced similar climatic histories with relatively minor superimposed catchment specific processes (Fig. 5a). Furthermore, the results from the factor analyses suggest that in addition to a 7-lake all proxy composite, a simple 7-lake composite of just the BSi (relative spring/summer warmth, e.g., Geirsdóttir et al., 2009b) and C/N (cold or erosional activity) can increase our understanding of the evolution of Holocene climate in Iceland and disentangle the catchment responses (C/N) to climate change and the temperature forcing (BSi). A comparison of such reconstructions with the composite of all proxies/all lakes is shown in Figure 5b and 5c.

15 **5 Discussion**

**5.1 The 4.2 ka event in Iceland and the role of volcanism**

The overall (first-order) trend of both the 7-lake all proxies and BSi composites track the orbital cycle of summer insolation across the Northern Hemisphere after peak insolation ~11 ka (Fig. 5b,c). Superimposed on the first-order decrease of composite BSi-inferred relative temperature are apparent step changes at ~5.5 ka, ~4.5 ka, ~3 ka and ~1.5 ka, with the lowest temperatures culminating during the LIA between 0.7 and 0.1 ka. However, the prominent step ~4.5-4.0 is the first distinct step that can be correlated between all seven lake records (Fig. 3b). This period spans the well-known ~4.2 ka event identified as a period of abrupt climate change elsewhere around the globe, manifested by pronounced dry conditions resulting in severe societal collapses in the Eastern Mediterranean (e.g. Weiss, 2016; 2017; Weiss et al., 1993; Cullen et al., 2000), and cooler and/or wetter conditions at higher altitudes in the Alps (Zanchetta et al., 2011, 2016). However, depending on the proxy and/or site chosen, the radiometric ages constraining these records from outside of Iceland indicate a much broader interval from 4.5 to 3.5 ka (Gasse, 2000; Wang et al., 2016; LeRoy et al., 2017; Railsback et al., 2018). Discerning whether the period of major climate perturbation in Iceland between 4.5-4.0 ka relates to the global expressions of the 4.2 ka climate event remains an open question.

Two steps toward cooling in the Icelandic lacustrine records, coincide with two of the largest explosive eruptions of the Holocene in Iceland; the Hekla 4 (3826+/-12 $^{14}$C BP/4197 cal yr BP; Dugmore et al., 1995), and Hekla 3 (2879+/-34 $^{14}$C BP/3006 cal yr BP; Dugmore et al., 1995). Both these Hekla eruptions deposited at least 1 cm of tephra over 80% of the surface of Iceland and are important tephrochronological markers throughout Europe, and importantly, a tie-point between most of our lake sediment records.. Because tephra fall-out from these eruptions likely disrupted local ecosystems (Larsen et al., 2011; Christensen, 2013; Eddudóttir et al., 2017), it is possible that they artificially imply colder conditions in Icelandic

Áslaug Geirsdóttir 12/5/2018 8:18 AM

Áslaug Geirsdóttir 12/18/2018 3:01 PM

Áslaug Geirsdóttir 12/5/2018 11:40 AM

Áslaug Geirsdóttir 12/3/2018 2:37 PM
**Moved (insertion) [1]**

Áslaug Geirsdóttir 12/18/2018 3:07 PM

Áslaug Geirsdóttir 12/5/2018 1:59 PM

lake sediment records from induced terrestrial erosion (higher C:N), which we would normally attribute to periods of cold and windy winters.. However, the Hekla 4 and Hekla 3 tephra are not found in either SKR or TRK (two lakes around Drangajökull in northwest Iceland), and thus the proxy records from these two lakes provide unequivocal evidence for cooling at these times unrelated to tephra-induced catchment disturbance or soil erosion (Harning et al., 2018a, b).

5    In terms of potential regional climate and temperature changes induced by the Hekla 4 eruption, the bulk of the tephra emitted was fine to very fine ash with residence time in the atmosphere presumably on the order of days to weeks, which together can lead to significant impact on a local to regional scale on time–scales ranging from weeks to ~3 years. The volume of the tephra is estimated to be 13.3 km$^3$ (= 1.8 km$^3$ when calculated as dense rock) deposited from a >30 km high eruption column, which makes it a category VEI5 event (Stevenson et al 2015). Although the aerosol cloud produced by the
10   Hekla 4 eruption had at least a hemispheric coverage, its impact on atmospheric properties and processes would have been negligible as data indicate that the Hekla 4 magma carried only about 20-70 ppm SO$_2$ (e.g. Portnyagin et al., 2012), compared to, for example the Laki eruption 1783-1784 in Iceland, which carried 1600 ppm (Thordarson et al., 1996). On the other hand, Hekla 4 magma would have carried between 500-1000 ppm Cl (Laki, 310 ppm) and 1300-1600 ppm F (Laki 660 ppm). Although the atmospheric venting of sulphur was negligible (<0.5 Tg), the halogen emissions would have been
15   significant or on the order of 10-15 Tg (Sverrisdóttir et al., 2007; Portnyagin et al., 2012; Stevenson et al., 2015). The halogens released by the Hekla 4 eruption into the stratosphere may thus have resulted in hemispheric depletion of ozone and hence stratospheric cooling on the same scale. However, the current unknown is how that cooling translates into surface cooling and associated changes in the tropospheric weather systems.

Although the origin and climatic impact of the 4.2 ka event remains poorly understood, it has previously been
20   linked to both volcanism (e.g. Antoniades et al., 2018) and ocean-atmospheric circulation changes in the North Atlantic (Bond et al., 2001; Bianchi and McCave, 1999; deMenocal, 2001). The similarity in the timing of cool events at higher latitudes in the Northern Hemisphere at this time further supports a temporary climate shift around the North Atlantic, most likely associated with millennial-scale variability in the AMOC and/or the sub-polar gyre (e.g., Risebrobakken et al., 2011; Thornalley et al., 2009; Trouet et al., 2009; Orme et al., 2018; Moreno-Chamarro et al., 2017; Zhong et al., 2018). Potential
25   contributions to local and/or hemispheric climate changes resulting from Hekla 4 magma degassing, however, remains a topic of further research.

**5.2 The demise and growth of the Icelandic glaciers during the Holocene**

Two of our lakes, HVT and TRK, are currently glacial lakes allowing their sediment records track the activity of Langjökull
30   and Drangajökull during periods when these ice caps attain sufficient dimensions to occupy the respective lake catchments (Fig. 2, 6). Hence, climate proxies in these lake sediments record the growth and demise of an upstream glacier (e.g., Briner et al., 2010; Larsen et al., 2012; Harning et al., 2018b). Both lake sediment records demonstrate that glacier ice disappeared from their catchments prior to 9 ka (Larsen et al., 2012; Harning et al., 2016b, 2018b). Furthermore, detailed sedimentological analyses of HVT indicate that the lake did not receive glacial meltwater between 7.9 and 5.5 ka (Larsen et

al., 2012). Glacier modeling experiments show that the demise of Langjökull and Drangajökull during the early Holocene required summer temperatures to rise ~3°C above the late 20[th] Century average at this time (Flowers et al., 2008; Anderson et al., 2018).

At 5.5 ka sedimentological analyses of HVT (increase in sediment accumulation rate, higher MS values and diminished biological productivity) show the first abrupt change toward cooler conditions and glacier occupation of the lake catchment (Larsen et al., 2011; Fig. 3a, b). The hydrologically coupled ice-sheet model employed to simulate the evolution of Langjökull through the Holocene also captures the inception of the modern ice cap prior to 5 ka (Flowers et al. 2008), concurrent with the first abrupt change in the BSi, MS and the sediment accumulation rate records (Fig. 6a). The second distinct change in HVT between 4.5 and 4.0 ka towards cooler climate is not only reflected in a suppressed algal productivity and increased landscape instability but a shift from weakly stratified to finely laminated sediments signaling the onset of a glacial-lacustrine-dominated HVT catchment. The Hekla 4 tephra layer generated from the explosive Hekla eruption dated to ~4.2 ka marks the culmination of this second change towards cooler climate that started around 4.5 ka. At 3 ka HVT developed the first distinctly varved sediment record, reflecting increased glacier activity in the catchment (Larsen et al., 2011). This again coincides with the eruption of the Hekla volcano at 3 ka and deposition of the Hekla 3 tephra. In both Hekla cases, the impact of the tephra in the HVT catchment has previously been linked to increased soil erosion as reflected in the C/N record (Larsen et al., 2011; 2012), which may obscure the impacts of global cooling related to the 4.2 ka event at the time of the Hekla 4 eruption. However, the potential impacts from Hekla 4 magma degassing may have contributed to the non-linear cooling step resulting in further growth of Langjökull under a background of decreasing summer insolation. However, the steepest decline in algal productivity (BSi in HVT) and the largest rate of glacier advance (Larsen et al., 2011) began later, between 1.8 and 1.5 ka and culminated during the LIA (Fig. 6a). This pattern is supported by the glacier numerical simulations, which suggest Langjökull attained its maximum volume during the Little Ice Age; first around 1840 CE and then around 1890 CE when estimated temperature were 1.5°C below the 1960-1990 average (Flowers et al., 2007) (Fig. 6c).

Unlike HVT, glacier ice did not reach Tröllkonuvatn's catchment until 1 ka, as illustrated by the lack of meltwater-derived clastic sediment in the lake between 9 and 1 ka (Harning et al., 2018b). However, Drangajökull was actively expanding into another threshold lake catchment on its current southeastern margin by at least 2.3 ka (Harning et al., 2016a). Further evidence for late Holocene advances of Drangajökull come from [14]C-dates on dead vegetation emerging from the currently receding northern and southern ice margins, which define the timing of persistent ice margin expansion at these locations (Harning et al., 2016a, 2018b). The threshold lake records combined with [14]C-dated emerging dead vegetation define five periods of increasing ice cap dimensions at ~2.3, 1.8, 1.4, 1 and 0.5 ka, where the final two ice margin advances (1 and 0.5 ka) are interpreted from Tröllkonuvatn's sediment record. These periods of ice growth are matched by decreases in BSi and increase in CN, suggesting regional climate change is the primary controlling factor (Harning et al., 2018b). The numerical simulations by Anderson et al. (2018) supports Drangajökull's late Holocene appearance, subsequent expansion and maximum dimensions between 0.5 and 0.3 ka with temperatures likely 1.0 to 1.2°C below the 1960-1990 reference

Áslaug Geirsdóttir 12/18/2018 3:32 PM

Áslaug Geirsdóttir 12/3/2018 3:05 PM
Deleted: Superimposed on the first-order decrease of BSi-inferred relative temperature are apparent step changes at ~5.5 ka, ~4.5 ka, ~3 ka and ~1.5 ka, with the lowest temperatures culminating during the LIA between 0.7 and 0.1 ka. Low BSi values coinciding with high and increasing C/N values at these steps suggest an increase in the proportion of terrestrial organic matter delivered to the lakes (soil erosion) relative to primary aquatic organic matter during cold times (Fig. 5b, c). The fastest rate of change occurs after 1.5 ka when the BSi values reflect the continuous decline in primary productivity at the same time that C/N values persistently increase, reflecting the catchment response to climate change (Fig. 5c, 6). ... [5]

Áslaug Geirsdóttir 12/9/2018 12:48 PM

Áslaug Geirsdóttir 12/9/2018 12:48 PM

Áslaug Geirsdóttir 12/9/2018 12:48 PM

Áslaug Geirsdóttir 12/5/2018 2:07 PM

Áslaug Geirsdóttir 12/18/2018 3:37 PM

Áslaug Geirsdóttir 12/18/2018 6:46 PM

Áslaug Geirsdóttir 12/18/2018 3:37 PM
Comment [1]: Something on the possible effect from the volcanic eruptions themselves – what is the effect of having fine ash in the atmosphere – could be local effect in Iceland etc…

Áslaug Geirsdóttir 12/18/2018 6:46 PM

Áslaug Geirsdóttir 12/5/2018 2:11 PM
Comment [2]: Something on the possible effect from the volcanic eruptions themselves – what is the effect of having fine ash in the atmosphere – could be local effect in Iceland etc…

Áslaug Geirsdóttir 12/18/2018 3:37 PM

Áslaug Geirsdóttir 12/5/2018 2:09 PM

Áslaug Geirsdóttir 12/5/2018 2:09 PM

Áslaug Geirsdóttir 12/18/2018 3:38 PM

temperature.

[revised manuscript text omitted]

**5.3 The "onset" of Neoglaciation**

Comparing the first indication of temperature lowering and glacier formation after the HTM in terrestrial regions in and around the northern North Atlantic reveals certain similarities and shows regionally consistent increases in millennial-
15  scale cooling rates. The monotonic Holocene decline of Northern Hemisphere summer insolation was most likely the primary driver for the first expansion of the cryosphere at ~5 ka identified in Baffin Island, eastern Greenland, in the highlands of Iceland, western Svalbard and in western Norway (Funder, 1978; Nesje et al., 2001; Jennings et al., 2002, Masson-Delmotte et al., 2005; Bakke et al., 2005a,b, 2010; Vinther et al., 2009; Larsen et al., 2012; Balascio et al. 2015; Solomina et al., 2015; Røthe et al., 2015, 2018; van der Bilt et al., 2015; Gjerde et al., 2016; Briner et al., 2016; Miller et al.,
20  2017). Although the gradual decline in summer insolation progressively lowered the ELA, the significant stepwise trend apparent in the Icelandic records and other records around the North Atlantic suggests that strong local to regional feedbacks modulated the primary insolation forcing. The rate of cryosphere expansion at 4.5-4.0 ka and particularly after 1.5 ka documents contemporaneous shifts in the northern North Atlantic region. Such episodic ice expansion cannot be explained by the summer insolation forcing alone and requires additional forcing or internal climate variability. Variations in the
25  strength of the thermohaline circulation, weakening of the northward heat transport of the AMOC and/or increasing influence of the Arctic waters influence all these locations. Changes in the strength of AMOC and/or the subpolar gyre and changes in the Arctic sea ice extent with the associated meridional heat transport into the Arctic have been related to past cooling events, particularly during the last 2 ka (Trouet et al., 2009, 2012; Lehner et al., 2013; Cabedo-Sanz et al., 2016; Moreno-Chamarro et al., 2017; Zhong et al., i2018).

30  The time interval known as "Neoglaciation" was defined by Porter and Denton (1967) as "the climatic episode characterized by rebirth and/or growth of glaciers following maximum shrinkage during the Hypsithermal [now HTM] interval". Porter (2000) noted that Neoglaciation is a geologic-climate unit based on physical geological evidence of glacier expansion, and that palynological evidence of climate change was excluded from the definition. The use of the phrase "onset of Neoglaciation" in the literature has been broader than the original definition and is commonly attributed to the first

Áslaug Geirsdóttir 12/4/2018 9:31 PM
**Moved down [2]:** Comparing the first indication of temperature lowering and glacier formation after the HTM in terrestrial regions in and around the northern North Atlantic reveals certain similarities and shows regionally consistent increases in millennial-scale cooling rates. The monotonic Holocene decline of Northern Hemisphere summer insolation was most likely the primary driver for the first expansion of the cryosphere at ~5 ka identified in Baffin Island, eastern Greenland, in the highlands of Iceland, western Svalbard and in western Norway (Funder, 1978; Nesje et al., 2001; Jennings et al., 2002, Masson-Delmotte et al., 2005; Bakke et al., 2005a,b, 2010; Vinther et al., 2009; Larsen et al., 2012; Balascio et al. 2015; Solomina et al., 2015; Røthe et al., 2015, 2018; van der Bilt et al., 2015; Gjerde et al., 2016; Briner et al., 2016; Miller et al., 2017;). Although the gradual decline in summer insolation progressively lowered the ELA, the significant stepwise trend in the Icelandic records suggests that strong local to regional feedbacks modulated the primary insolation forcing. The rate of cryosphere expansion at 4.5-4.0 ka and particularly after 1.5 ka documents contemporaneous shifts in the northern North Atlantic region. Such episodic ice expansion cannot be explained by the summer insolation forcing alone and requires additional forcing or internal climate variability. Variations in the strength of the thermohaline circulation, weakening of the northward heat transport of the AMOC and/or increasing influence of the Arctic waters influence all these locations. Changes in the strength of AMOC and/or the subpolar gyre and changes in the Arctic sea ice extent with the associated meridional heat transport into the Arctic have been related to past cooling events, particularly during the last 2 ka (Trouet et al., 2009, 2012; Lehner et al., 2013; Moreno-Chamarro et al., 2017; Zhong et al., in review), supporting our conclusions .

Áslaug Geirsdóttir 12/4/2018 9:31 PM

Áslaug Geirsdóttir 12/4/2018 9:31 PM
**Moved (insertion) [2]**

Áslaug Geirsdóttir 12/5/2018 2:18 PM

Áslaug Geirsdóttir 12/9/2018 3:37 PM

Áslaug Geirsdóttir 12/9/2018 3:37 PM

Áslaug Geirsdóttir 12/4/2018 9:31 PM

apparent indication of lowered temperature or increased rate of summer cooling rather than simply the renucleation and/or expansion of glaciers.

Following Porter's (2000) definition, the onset of Neoglaciation in Iceland based on our lake records occurred before 5.0 ka for Langjökull, despite the initial growth of Drangajökull occurring much later, at ~2.3 ka. This indicates that the spatio-temporal nucleation of glaciers in Iceland was asynchronous and likely reflects the relation between the regional ELA (primarily controlled by summer temperature) and topography. The nature of the topography (i.e., the hypsometry) controls how quickly the glacier will expand after the ELA intersects the topography. Because the topography under Drangajökull is a plateau-like landscape with a large area at its highest elevation, Drangajökull will grow quickly once the ELA intercepts the topography (Anderson et al., 2018).

The current ELA pattern reflects the patterns of temperature and precipitation across Iceland. Temperature differences from the south to the northwest reflect the prevailing wind direction and proximal ocean surface temperatures. Precipitation also impacts ELAs across Iceland but varies primarily due to the interaction of local topography with prevailing winds (Crochet et al., 2007). Because Icelandic glaciers are most sensitive to temperature, we expect the timing of glacier inception to be controlled by the rate at which Holocene temperatures decline following the HTM and the subglacial topographic setting of each of the current Icelandic ice caps.

Changes in the ELA at present are most sensitive to summer temperature, and assuming the same holds for most of the Holocene, we apply a simple approach to define the "onset" of Neoglaciation in Iceland by comparing the Holocene evolution of BSi (a measure of variations in summer temperature) and C/N (a measure of variations in catchment stability, which independently tracks summer temperature, modulated by volcanism) in the coastal lakes (SKR, TRK, TORF, HAK) and the highland lakes (HVT, ARN) (Fig. 7). Although the multi-proxy lake records document complex changes in terrestrial climate and glacier fluctuations in Iceland during the mid-to-late Holocene, coherent patterns of change are apparent. Based on glacier nucleation, our records show that initiation of Neoglacial cooling took place in the highlands of Iceland (Langjökull) around 5.5 ka, where the current ELA is ~1170 m (Fig. 6a, 7). This cooling is also mirrored by the retreat of woodland from 6000 to 4000 cal yr BP, reflected by decreased *Betula pubescens* pollen counts in a lake record from the northwest highlands (Eddudóttir et al., 2016). The rate of glacier growth likely increased between 4.5 ka and 4.0 ka when NE Vatnajökull (ELA 1320 m) nucleated (Fig. 6b), and continued near ~2.5 ka, when Drangajökull nucleated (ELA 675 m) (Fig. 6c). Consistent with this scenario is the distinct first-order cooling trend apparent in all seven lakes, including the combined highland (HVT, ARN) and the coastal (SKR, TRK, TORF, HAK) lake composites (Fig. 7). The proxies in the highland lakes, HVT and ARN, show a more abrupt and greater response to the temperature lowering, most likely due to a greater impact from catchment specific processes (tephra deposition and/or glacial activity), whereas the coastal lakes show a more subdued response, likely reflecting the moderating effect from SSTs (Fig.7). These lakes show not only striking similarities with the diatom-based SST record from the shelf north of Iceland (Jiang et al., 2015) but also with the IP$_{25}$-based sea-ice reconstruction (Cabedo-Sanz et al., 2016) and the ice-rafted debris (IRD) record based on quartz grain counts (Moros et al., 2006) (Fig. 7), both off the north coast of Iceland. The IP$_{25}$-based sea ice reconstruction shows a rise at ~5 ka from a

background state beginning earlier at 8 ka, and intensifies after 4.5 ka, broadly in line with the decreased abundance of planktic diatoms and lowering of SST (Jiang et al., 2015) (Fig.7). Further increases in drift ice were evident during the late Holocene after ~3.3 ka, with maximum sea ice after ~1.0 ka and during the Little Ice Age (Moros et al., 2006; Cabedo-Sanz et al., 2016). The intensification around 4.5 ka seen in our lake records is in line with increased strength of cold and fresh Polar Water via the East Greenland Current (EGC, Fig. 1) at 4.5 ka in the northern North Atlantic inferred from the decreased abundance of planktic foraminifera (e.g., Andersen et al., 2004; Jennings et al., 2011; Ólafsdóttir et al., 2010; Kristjánsdóttir et al., 2016; Perner et al., 2015, 2016).

A diatom-based SST reconstruction from a core retrieved from the Iceland Basin south of Iceland shows pronounced SST cooling between 4 and 2 ka, with warmer temperatures prior to 4 ka but also between 2 and 1.5 ka (Orme et al., 2018). Orme et al. (2018) primarily explain the cool interval between 4 and 2 ka as a response to a persistently negative mode of the North Atlantic Oscillation (NAO) that caused strengthening of the northerly winds east of Greenland, which in turn strengthened the East Greenland Current, bringing cool Arctic water as far south as the Icelandic Basin. The abrupt increases in IP$_{25}$ at ~1.5 and 0.7 ka are coincident with increased rate of cooling identified in the Icelandic lacustrine temperature record, suggesting significant coupling between the marine and terrestrial systems in Icelandic waters. From this treatment, we conclude that the primary driver explaining changes in our climate proxies was the decline in summer insolation modulated by changes in ocean circulation and associated sea surface temperature (SST) around the coast of Iceland. Explosive Icelandic volcanism whether or not it caused the temperature decline produced important perturbations to lake catchments through tephra deposition, often resulting in the recovery of catchment systems to a different equilibrium state.

**6 Conclusion**

- The results from factor analyses of 6 climate proxies from 7 Icelandic lake sediment records suggest that a simple composite of BSi (relative spring/summer temperature) and C/N (cold or erosional activity) serve to increase our understanding of the evolution of Holocene climate in Iceland and disentangle the catchment responses to climate change and the temperature forcing.
- The Holocene thermal maximum was warm enough to result in a mostly ice-free Iceland by 9 ka. The regular decline of summer insolation after 11 ka, plausibly amplified by responses elsewhere in the North Atlantic region, led to decreases in summer temperature and the destabilization of catchments in non-glacial lakes.
- Based on our seven lake records, the onset of Neoglaciation is registered in the highlands of Iceland (Langjökull) around 5.5 ka, when its ELA intercepted the local subglacial topography. The delayed nucleation of northeast Vatnajökull (4.4 ka) and DRangajökull (2.3 ka) can be explained by lower topographic thresholds and the necessity of

Áslaug Geirsdóttir 12/5/2018 2:26 PM

Áslaug Geirsdóttir 12/5/2018 2:26 PM

Áslaug Geirsdóttir 12/5/2018 2:26 PM

Áslaug Geirsdóttir 12/18/2018 6:10 PM

Áslaug Geirsdóttir 12/18/2018 6:10 PM

Áslaug Geirsdóttir 12/18/2018 6:10 PM

Áslaug Geirsdóttir 12/18/2018 6:13 PM

lower summer temperatures for regional ELAs to intercept their respective topography. Our observed glacier nucleations all coincide with the stepped cooling reflected in all 7 lake sediment records.

- The results from the factor analyses suggest that a simple composite of just the BSi as a measure of change in relative spring/summer warmth and C/N as a measure of cold or erosional activity could, along with the composites, serve to increase our understanding of the evolution of Holocene climate in Iceland and disentangle the catchment responses to climate change and the temperature forcing

- Episodic glacier expansion between 4.5 and 4.0 ka cannot be explained by the summer insolation forcing alone and require additional forcings, likely linked to ocean circulation and/or expansion of Arctic Ocean sea ice or internal climate variability. Magma degassing from local Icelandic eruptions may have also contributed but requires further research to confirm

- The prominent step toward cooling at 4.5-4.0 ka is statistically indistinguishable from the global ~4.2 ka event, and also coincides with Hekla 4, one of the largest explosive eruptions of the Holocene in Iceland. Although the ash generated from the eruption likely disrupted local catchment stability in Iceland, its impact on global atmospheric properties and processes would have been negligible due to the low sulphur content of the eruption.

- North Atlantic circulation features (NAO and AMOC) likely influenced the temperature decline in Iceland during the late Holocene, modulating the hemispherically symmetric forcings (insolation, irradiance and volcanism).

**7 Author contribution**

ÁG wrote the manuscript and performed statistical analysis with JTA. ÁG and GHM provided funding for the research reported here through the Icelandic RANNIS and US NSF funding agencies. DJH, LA, CF, DJL, TT provided support on the interpretation and gave their comments and agreement during the writing process.

**8 Competing interests**

The authors declare that they have no conflict of interest.

**9 Acknowledgements**

This work was supported primarily by the Icelandic Center for Research through grants awarded to ÁG and GHM (#130775051 and Grant-of-Excellence #141573052) and several grants awarded to ÁG from the UI Research Fund. We thank Sædís Ólafsdóttir, Celene Blair, Sarah Crump, Sydney Gunnarson, Thorsteinn Jónsson and Sveinbjörn Steinthorsson, who all contributed to this work by either taking part in field work activities, laboratory analyses and/or discussion. Thorough and constructive reviews from two anonymous reviewers have substantially improved the manuscript.

**Table 1: Lake settings.**

| Lake name | Elevation (m asl) | Latitude | Longitude | Lake catch-ment (km$^2$) | Lake area (km$^2$) | Depth max (m) | References |
|---|---|---|---|---|---|---|---|
| Vestra Gíslholtsvatn (VGHV) | 61 | 63°56′33N | 20°31′08W | 4.2 | 1.57 | 15 | Blair et al., 2015 |
| Hvítárvatn (HVT) | 422 | 64°38′13N | 19°51′29W | 820 | 28.9 | 83 | Larsen et al., 2011, 2012 |
| Arnarvatn Stóra (ARN) | 540 | 64°57′04N | 20°20′40W | 61 | 4 | 3 | Gunnarson, 2017 |
| Torfdalsvatn (TORF) | 52 | 66°03′38N | 20°22′59W | 2.76 | 0.4 | 5.8 | Florian, 2016 |
| Haukadalsvatn (HAK) | 32 | 65°03′02N | 21°38′15W | 172 | 3.3 | 42 | Geirsdóttir et al., 2009, 2013 |
| Tröllkonuvatn (TRK) | 366 | 66°08′39N | 22°03′33W | 9.4 | 1.3 | 16.4 | Harning et al., 2016, 2018 |
| Skorarvatn (SKR) | 183 | 66°15′22N | 22°1931W | 1.2 | 0.2 | 25 | Harning et al., 2016, 2018 |

Áslaug Geirsdóttir 12/18/2018 6:36 PM

**Table 2: Factor analyses**

| Object label | Factor 1 | Factor 2 | Factor 3 | Communality | Unique | Varimax Factor 1 | Varimax Factor 2 | Varimax Factor 3 |
|---|---|---|---|---|---|---|---|---|
| Eigenvalue | 3.4128 | 1.1684 | 0.571 | | | | | |
| Variance (%) | 56.8808 | 19.4734 | 9.5172 | | | | | |
| Variance (cum.%) | 56.8808 | 76.3542 | 85.8714 | | | | | |
| | | | | | | 1 | 2 | 3 |
| **MS** | 0.7936 | 0.1134 | 0.538 | 0.9321 | 0.0679 | 0.2624 | 0.1081 | 0.9228 |
| **C/N** | 0.0525 | 0.9686 | -0.0584 | 0.9444 | 0.0556 | 0.0062 | 0.9717 | 0.0146 |
| **TC** | -0.8616 | 0.2605 | -0.2196 | 0.8584 | 0.1416 | -0.5439 | 0.2405 | -0.7105 |
| **BSi** | -0.7982 | 0.2665 | 0.3827 | 0.8546 | 0.1454 | -0.8769 | 0.2083 | -0.2056 |
| **δ13C** | -0.8621 | -0.2683 | 0.103 | 0.8258 | 0.1742 | -0.7097 | -0.3086 | -0.4764 |
| **δ15N** | 0.811 | 0.0801 | -0.27 | 0.737 | 0.263 | 0.7899 | 0.1302 | 0.3101 |

[Figure]

Figure 1. Location of Iceland in the northern North Atlantic and relative to the major ocean surface currents. EGC = East Greenland Current; EIC = East Iceland Current; NIIC = North Icelandic Irminger Current; IC = Irminger Current; NAC = North Atlantic Current.

Áslaug Geirsdóttir 12/18/2018 6:39 PM

[Figure]

Figure 2. Air photographs of the seven lakes: a. Tröllkonuvatn TRK, b. Skorarvatn (SKR), c. Torfdalsvatn TORF, d. Haukadalsvatn HAK, e. Vestra Gíslholtsvatn VGHV, f. Arnarvatn Stóra ARN, g. Hvítárvatn HVT. Air photographs (2005) are high resolution (0.5 m) from Loftmyndir.

[Figure]

Figure 3a. Comparison of normalized composite proxy records of the seven lakes. Hekla 4 (H4; 4.2 ka) and Hekla 3 (H3; 3.0 ka) are marked with orange coloured hatched lines. Blue hatched lines mark steps in the records at 5.5 ka, 4.5 ka and 1.5 ka.

[Figure]

Figure 3b. Comparison of normalized composite BSi records of the seven lakes. Hekla 4 (H4; 4.2 ka) and Hekla 3 (H3; 3.0 ka) are marked with orange coloured hatched lines. Blue hatched lines mark steps in the records at 5.5 ka, 4.5 ka and 1.5 ka.

[Figure]

Figure 4. Factor 1 and 2 explain 76.4%. See Table 2 for details.

[Figure]

Figure 5. a. Composite records of all seven lakes combined compared to the 2-lake composite of Geirsdóttir et al. (2013), b. BSi composite of all 7 lakes compared to composite of all proxies in 7 lakes, c. BSi all lakes and C/N all lakes. Dashed lines mark the step changes identified.

[Figure]

Figure 6. a. BSi and C/N from HVT and Lögurinn. b. BSi and C/N from Lögurinn, c. BSi from TRK. Blue shaded bars indicates glacier inception at each lake catchment.

Áslaug Geirsdóttir 12/18/2018 6:38 PM

[Figure]

**Figure 7.** Composite record from all seven lakes compared to **a.** 65°N summer insolation (Berger and Loutre, 1991; **b.** Volcanic Forcing W/m2 (Kobashi et al., 2017), **c.** MD99-2275 diatom based SST (Jiang et al., 2015); **d.** Renland-Agassiz ice core record (normalized d18O record and its variance) from Greenland (Vinther et al., 2009). **e.** Quartz (normalized) from MD99-2269 as an indication of sea ice in the northern North Atlantic (Moros et al., 2006), **f.** IP25 record of sea ice (Capedo-Sanz et al., 2016) **g.** BSi coastal lakes SKR, TRK, TORF, and HAK, **h.** BSi highland lakes HVT+ARN, **i.** BSi all lakes.

Blue hatched lines show the 5.5, 4.5 and 1.5 ka steps in the Icelandic lake records. Orange hatched lines show the timing of Hekla 4 and Hekla 3 volcanic eruptions.

[Figure]

| Page 7: [1] Deleted | Áslaug Geirsdóttir | 12/18/18 3:01 PM |
|---|---|---|

the

| Page 7: [1] Deleted | Áslaug Geirsdóttir | 12/18/18 3:01 PM |
|---|---|---|

the

| Page 7: [1] Deleted | Áslaug Geirsdóttir | 12/18/18 3:01 PM |
|---|---|---|

the

| Page 7: [1] Deleted | Áslaug Geirsdóttir | 12/18/18 3:01 PM |
|---|---|---|

the

| Page 7: [1] Deleted | Áslaug Geirsdóttir | 12/18/18 3:01 PM |
|---|---|---|

the

| Page 7: [1] Deleted | Áslaug Geirsdóttir | 12/18/18 3:01 PM |
|---|---|---|

the

| Page 7: [1] Deleted | Áslaug Geirsdóttir | 12/18/18 3:01 PM |
|---|---|---|

the

| Page 7: [1] Deleted | Áslaug Geirsdóttir | 12/18/18 3:01 PM |
|---|---|---|

the

| Page 7: [1] Deleted | Áslaug Geirsdóttir | 12/18/18 3:01 PM |
|---|---|---|

the

| Page 7: [1] Deleted | Áslaug Geirsdóttir | 12/18/18 3:01 PM |
|---|---|---|

the

| Page 7: [1] Deleted | Áslaug Geirsdóttir | 12/18/18 3:01 PM |
|---|---|---|

the

| Page 7: [2] Deleted | Áslaug Geirsdóttir | 12/18/18 3:07 PM |
|---|---|---|

reflect

| Page 7: [2] Deleted | Áslaug Geirsdóttir | 12/18/18 3:07 PM |
|---|---|---|

reflect

| Page 7: [2] Deleted | Áslaug Geirsdóttir | 12/18/18 3:07 PM |
|---|---|---|

reflect

| Page 7: [2] Deleted | Áslaug Geirsdóttir | 12/18/18 3:07 PM |
|---|---|---|

reflect

| Page 7: [2] Deleted | Áslaug Geirsdóttir | 12/18/18 3:07 PM |
|---|---|---|

reflect

| Page 7: [2] Deleted | Áslaug Geirsdóttir | 12/18/18 3:07 PM |
|---|---|---|

reflect

| Page 7: [2] Deleted | Áslaug Geirsdóttir | 12/18/18 3:07 PM |
|---|---|---|

reflect

| Page 7: [2] Deleted | Áslaug Geirsdóttir | 12/18/18 3:07 PM |
|---|---|---|

reflect

| Page 7: [2] Deleted | Áslaug Geirsdóttir | 12/18/18 3:07 PM |
|---|---|---|

reflect

| Page 7: [3] Deleted | Áslaug Geirsdóttir | 12/5/18 1:59 PM |
|---|---|---|

Two of the more prominent

| Page 7: [3] Deleted | Áslaug Geirsdóttir | 12/5/18 1:59 PM |
|---|---|---|

Two of the more prominent

| Page 7: [3] Deleted | Áslaug Geirsdóttir | 12/5/18 1:59 PM |
|---|---|---|

Two of the more prominent

| Page 7: [3] Deleted | Áslaug Geirsdóttir | 12/5/18 1:59 PM |
|---|---|---|

Two of the more prominent

| Page 8: [4] Deleted | Áslaug Geirsdóttir | 12/5/18 12:13 PM |
|---|---|---|

The overall (first-order) trend of the BSi composite reflects the orbital cycle of summer insolation across the Northern Hemisphere after peak insolation ~11 ka (Fig. 5b,c). The presence of the 10 ka Grímsvötn tephra series (including the so-called Saksunarvatn tephra) in all of the seven lakes and other high-elevation sites across much of Iceland (Stötter et al., 1999; Caseldine et al., 2003; Jóhannsdóttir, 2007; Geirsdóttir et al., 2009a; Larsen et al., 2012; Harning et al., 2018a) demonstrates that the highlands were mostly ice-free by the time of the eruption ~10.3 ka, with most of the Iceland Ice Sheet gone from the interior before ~9 ka (Larsen et al., 2012, Harning et al., 2016a, 2018b; Striberger et al., 2012; Gunnarson, 2017).

| Page 9: [5] Deleted | Áslaug Geirsdóttir | 12/3/18 3:05 PM |
|---|---|---|

Superimposed on the first-order decrease of BSi-inferred relative temperature are apparent step changes at ~5.5 ka, ~4.5 ka, ~3 ka and ~1.5 ka, with the lowest temperatures culminating during the LIA between 0.7 and 0.1 ka. Low BSi values coinciding with high and increasing C/N values at these steps suggest an increase in the proportion of terrestrial organic matter delivered to the lakes (soil erosion) relative to primary aquatic organic matter during cold times (Fig. 5b, c). The fastest rate of change occurs after 1.5 ka when the BSi values reflect the continuous decline in primary productivity at the same time that C/N values persistently increase, reflecting the catchment response to climate change (Fig. 5c, 6).

between 0.7 and 0.1 ka. Low BSi values coinciding with high and increasing C/N values at these steps suggest an increase in the proportion of terrestrial organic matter delivered to the lakes (soil erosion) relative to primary aquatic organic matter during cold times (Fig. 5b, c). The fastest rate of change occurs after 1.5 ka when the BSi values reflect the continuous decline in primary productivity at the same time that C/N values persistently increase, reflecting the catchment response to climate change (Fig. 5c, 6).

| Page 9: [5] Deleted | Áslaug Geirsdóttir | 12/3/18 3:05 PM |
|---|---|---|

Superimposed on the first-order decrease of BSi-inferred relative temperature are apparent step changes at ~5.5 ka, ~4.5 ka, ~3 ka and ~1.5 ka, with the lowest temperatures culminating during the LIA between 0.7 and 0.1 ka. Low BSi values coinciding with high and increasing C/N values at these steps suggest an increase in the proportion of terrestrial organic matter delivered to the lakes (soil erosion) relative to primary aquatic organic matter during cold times (Fig. 5b, c). The fastest rate of change occurs after 1.5 ka when the BSi values reflect the continuous decline in primary productivity at the same time that C/N values persistently increase, reflecting the catchment response to climate change (Fig. 5c, 6).

| Page 9: [6] Deleted | Áslaug Geirsdóttir | 12/5/18 2:07 PM |
|---|---|---|

at 4.5

| Page 9: [7] Deleted | Áslaug Geirsdóttir | 12/18/18 3:37 PM |
|---|---|---|

on the landscape in either case is unambiguous and may thus be one representation of amplified catchment response to the declining summer insolation and contemporaneous cooling that may have resulted in further growth of Langjökull[1]. The fastest rate of decline in algal productivity (BSi in HVT)

and the most apparent glacier advance began between 1.8 and 1.5 ka culminating in the LIA (Fig. 6a). This pattern is supported by the numerical simulations, which suggest Langjökull attained its maximum volume during the Little Ice Age; first around 1840 CE and then around 1890 CE when estimated temperature were 1.5°C below the 1960-1990 average (Flowers et al., 2007) (Fig. 6c).

| Page 9: [8] Deleted | Áslaug Geirsdóttir | 12/18/18 3:38 PM |
|---|---|---|

the

| Page 9: [8] Deleted | Áslaug Geirsdóttir | 12/18/18 3:38 PM |
|---|---|---|

the

| Page 9: [8] Deleted | Áslaug Geirsdóttir | 12/18/18 3:38 PM |
|---|---|---|

the

| Page 9: [8] Deleted | Áslaug Geirsdóttir | 12/18/18 3:38 PM |
|---|---|---|

the

| Page 9: [8] Deleted | Áslaug Geirsdóttir | 12/18/18 3:38 PM |
|---|---|---|

the

| Page 13: [9] Deleted | Áslaug Geirsdóttir | 12/5/18 2:26 PM |
|---|---|---|